# Rapid and widespread white matter plasticity during an intensive reading intervention

Elizabeth Huber[1,2], Patrick M. Donnelly [1,2], Ariel Rokem [3] & Jason D. Yeatman [1,2]

White matter tissue properties are known to correlate with performance across domains ranging from reading to math, to executive function. Here, we use a longitudinal intervention design to examine experience-dependent growth in reading skills and white matter in grade school-aged, struggling readers. Diffusion MRI data were collected at regular intervals during an 8-week, intensive reading intervention. These measurements reveal large-scale changes throughout a collection of white matter tracts, in concert with growth in reading skill. Additionally, we identify tracts whose properties predict reading skill but remain fixed throughout the intervention, suggesting that some anatomical properties stably predict the ease with which a child learns to read, while others dynamically reflect the effects of experience. These results underscore the importance of considering recent experience when interpreting cross-sectional anatomy–behavior correlations. Widespread changes throughout the white matter may be a hallmark of rapid plasticity associated with an intensive learning experience.

[1] Institute for Learning and Brain Sciences, University of Washington, Seattle, WA 98195, USA. [2] Department of Speech and Hearing Sciences, University of Washington, Seattle, WA 98195, USA. [3] eScience Institute, University of Washington, Seattle, WA 98195, USA. Correspondence and requests for materials should be addressed to E.H. (email: ehuber@uw.edu) or to J.D.Y. (email: jyeatman@uw.edu)

Skilled reading requires orchestration of a large cortical network, and individual differences in reading performance have been linked to the properties of white matter tracts connecting portions of this network specialized for processing visual, acoustic, and semantic features[1–5]. Although individual differences in white matter are thought to reflect the joint influence of genetics and experience[6–8], white matter properties are often held to underlie variation in performance and to causally influence individual learning trajectories[9–11]. A number of recent studies, working within this framework, have identified features of the white matter that predict reading outcomes in dyslexia[12] and reading-related skills, like phonological awareness, in pre-reading children[10,13]. The implication of these observations is that underlying anatomical differences may predestine certain individuals to struggle with learning to read. In this view, differences in white matter properties could be considered a reflection of intrinsic deficits, which might be relatively resistant to remediation, but which could plausibly be used for early identification of individuals in need of extra educational support.

Successfully relating anatomical differences with behavioral outcomes requires an understanding of the timescale over which white matter tissue properties exhibit experience-dependent change and the anatomical specificity of these effects. White matter plasticity, including activity-dependent myelination and oligodendrocyte proliferation, has been observed in animal models over the timescale of days to weeks[14–16], and these effects coincide with changes in tissue properties measured non-invasively using diffusion-weighted magnetic resonance imaging (dMRI) in animals[17,18]. It has further been suggested that myelination may play a causal role in skill learning, since blocking the production of new myelinating oligodendrocytes inhibits motor skill development in mice[19], implying that changes in white matter are critical to the learning process, rather than epiphenomenal. It is not clear whether similar effects occur in the context of human learning, particularly for a complex skill like reading, which is typically acquired with many hours of practice over a large developmental window. However, the studies cited above strongly suggest that learning should be accompanied by rapid, measurable changes in white matter. Further, a number of recent studies highlight the surprising malleability of human white matter in response to short-term training[20–22], including training of reading and related skills[23–25]. This opens the possibility that correlations between white matter properties and behavior arise as temporary states within a highly plastic system that flexibly adapts to environmental demands. In this case, observed relationships between anatomy and behavior might be less stable than often presumed, given an appropriate change to the educational environment.

Here we test whether controlled changes to a child's educational environment induce changes in white matter tissue properties over the timescale of weeks. Using a longitudinal intervention design, we track improvements in reading skills, and accompanying changes in white matter, in a group of grade school-aged, struggling readers during 8 weeks of intensive (4 h each day, 5 days a week), one-on-one training in reading skills. We first examine learning effects within three tracts thought to carry signals critical for skilled reading[1–5,10,11,26–34]: the left arcuate fasciculus (AF), left inferior longitudinal fasciculus (ILF), and posterior callosal connections (CC). These pathways connect canonical reading-related regions within the ventral occipito-temporal (including the visual word form area (VWFA)[35–40]), superior temporal[41,42], and inferior frontal cortex[43], and hence, these tracts are considered to be part of the core circuitry for reading[27,39,40]. We find that the AF and ILF exhibit experience-dependent change within weeks of the intervention onset, while tissue properties within the posterior CC remain fixed. Moreover,

we illustrate the ambiguity of brain–behavior correlations measured in a dynamic system: As training rapidly alters an individual's white matter and behavior, cross-sectional correlations between white matter properties and reading skills change substantially between measurement sessions. Meanwhile, CC white matter properties, which do not change during training, remain correlated with reading skill throughout the intervention. We therefore suggest that some anatomical properties may be stable predictors of the ease with which a child learns to read, while others dynamically reflect the effects of experience. These effects likely arise from distinct mechanism that cannot be distinguished by cross-sectional studies. Finally, we test the hypothesis that experience-dependent plasticity is anatomically localized to specific tracts. Contrary to this anatomical-specificity hypothesis, we find that educational experience alters a widespread system of white matter tracts in concert with reading skills. This system includes, but is not limited to, the core reading circuitry.

## Results

**White matter and reading are correlated before intervention.** We began by replicating previously reported correlations between reading skill and properties of the white matter tracts connecting key components of the reading circuitry[1–5,10,11,26–28,32]. To summarize individual differences in reading, we report Reading Skill, a composite score that incorporates our full battery of reading tests from the Woodcock–Johnson[44] and Test of Word Reading Efficiency (TOWRE)[45] standardized assessments (see Methods for details, and Supplementary Fig. 1A). To characterize the cross-sectional relationship between white matter and reading, we calculated simple, bivariate correlations between Reading Skill and each diffusion metric at the pre-intervention baseline session. As shown in Fig. 1, pre-intervention (Session 1) measurements replicate previously reported correlations between reading scores and diffusion properties in the left arcuate, left ILF, and the CC: correlations between MD and Reading Skill are positive both in the intervention group and in the full sample containing intervention and control subjects (Fig. 1). The Reading Skill composite is a weighted sum of the individual reading tests, and similar effects are observed when examining correlations with the Woodcock–Johnson and TOWRE measures. Mirroring these effects, correlations between FA and reading are negative (Supplementary Fig. 2). While several previous studies report a negative relationship between FA and reading in these pathways[2,4,28], others report a positive relationship between FA and reading[1,26,32,46,47]. Thus, while properties of these pathways have consistently been shown to correlate with reading skill, the direction of this relationship is not consistent across studies or tracts (see ref. [48]). These inconsistencies may depend on factors like age, education, or socioeconomic status or may reflect the inherent ambiguity of dMRI metrics like FA, which can be influenced by a number of underlying biological phenomena with distinct, and potentially opposing, relationships to reading[28].

In addition to the tracts chosen a priori for analysis, we examined several other tracts previously shown to correlate with reading scores, albeit less consistently across studies[5,49], in a subsequent exploratory analysis. As shown in Supplementary Table 1, the left inferior frontal occipital fasciclus (IFOF) was also significantly correlated with reading skill (Bonferroni-corrected $p < 0.05$), and a number of other tracts showed moderate, non-significant correlations. Finally, to test whether correlations were specific to Reading Skill, as opposed to general academic ability, we calculated correlations with math scores (Woodcock–Johnson Calculation and Math Facts Fluency) and found that none of the tracts that significantly correlated with reading (including the AF, ILF, and CC) correlated with math skills. Indeed, neither MD nor

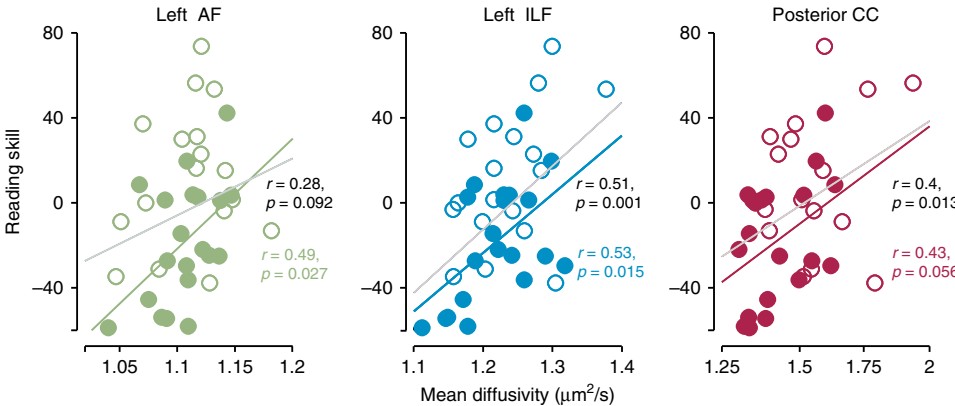

**Fig. 1** Pathways connecting the core reading circuitry correlate with pre-intervention Reading Skill. Tract average mean diffusivity (MD) is plotted as a function of pre-intervention (Session 1) reading skill. Best-fit lines plotted in gray give estimates for the full dataset, while colored lines show fits for the intervention subjects, alone. Correlations for the intervention subjects are given in colored text below the value estimated for the full dataset (in black)

FA showed a significant relationship to math skills in any of the tracts chosen for analysis.

**Intervention changes reading skill and white matter**. Reading skills improved substantially during the 8-week intervention period. Standard scores on the Woodcock–Johnson Basic Reading Composite, an untimed measure of reading accuracy, improved significantly over the course of the intervention ($F(1,77) = 59.75$, $p < 10^{-10}$, linear mixed effect model with a fixed effect of intervention time, in hours, and a random effect of subject). After 8 weeks, the intervention-group mean was within one standard deviation of the population norm ($100 +/- 15$): pre- vs. post-intervention scores were $80.00 +/- 3.50$ vs. $92.94 +/- 2.50$. In line with these results, scores on the TOWRE Index, a timed measure of reading, improved substantially ($F(1,77) = 53.69$, $p < 10^{-9}$), as did scores on the Woodcock–Johnson Reading Fluency subtest ($F(1,76) = 36.042$, $p < 10^{-7}$). In contrast, we found no evidence for change in math skills during the intervention (Woodcock–Johnson Calculation Score, $F(1,63) = 2.54$, $p = 0.12$; Woodcock–Johnson Math Fact Fluency: $F(68) = 1.87$, $p = 0.18$), confirming that the intervention specifically affected reading skills. Additional details of these analyses are given in Supplementary Fig. 1.

Growth in reading accuracy was specific to the intervention group, as indicated by a significant group (intervention vs. control) by time (days) interaction for the Woodcock–Johnson Basic Reading Composite: We saw a significant effect of group ($F(1,124) = 9.59$, $p = 0.0024$) but not time ($F(1,124) = 1.86$, $p = 0.17$) and a significant group-by-time interaction ($F(1,124) = 8.64$, $p = 0.0039$). For this analysis, we substituted "days" for "intervention hours," to provide a meaningful index of time for both the intervention and control groups. For intervention subjects, "days" were highly correlated with "intervention hours," since testing sessions were scheduled at regularly spaced intervals ($r(78) = 0.95$, $p < 001$). In the full control sample ($n = 19$), performance improved moderately with repeated testing for the timed measures (TOWRE and Reading Fluency), and thus we did not detect a significant group-by-time interaction for these tests, as shown in Supplementary Table 2. We attribute this result to practice effects among the most skilled readers. Indeed, in a reading-skill-matched subset of the control subjects ($n = 9$), we detected no change in reading scores over time and significant group-by-time interactions for all of the reading-related measures. In other words, skilled readers benefited slightly from repeated practice with the timed reading tests, while poor readers did not show any improvements with practice and only showed an improvement in performance as a result of the intervention program. All results for the reading-matched control group are shown in Supplementary Table 2, alongside results for the full non-intervention control group.

To test whether changes in reading skill were accompanied by measurable changes in white matter structure, we first examined MD and FA as a function of intervention time (hours) within the set of white matter tracts considered to be crucial for skilled reading[1–5,10,11,26–28,32] and which showed significant relationships with pre-intervention reading skill in the current sample: the left AF, left ILF, and posterior CC. Intervention-driven tissue changes were evident within the AF and ILF but not within the CC: Specifically, mean diffusivity (MD) decreased as a function of intervention hours within the left AF ($F(1,77) = 8.46$, $p = 0.0047$, linear mixed effect model with a fixed effect of intervention time, in hours, and a random effect of subject) and the left ILF ($F(1,77) = 7.28$, $p = 0.0086$), but not within the CC ($F(1,77) = 2.37$, $p = 0.13$). Subject motion did not change over time (Supplementary Fig. 3) and including subject motion as a covariate in the model did not change the results: MD decreased as a function of intervention hours within the left AF ($F(1,76) = 10.48$, $p = 0.0018$) and the left ILF ($F(1,76) = 9.53$, $p = 0.0028$), but not within the CC ($F(1,76) = 2.11$, $p = 0.15$). The decline in MD was accompanied by a linear increase in fractional anisotropy (FA) in the left AF ($F(1,76) = 3.98$, $p = 0.050$, fixed effect of intervention hours and a random effect of subject, with subject motion included as a covariate, as above) and the left ILF ($F(1,76) = 8.82$, $p = 0.0040$) but not in the CC ($F(1,76) = 0.24$, $p = 0.62$). Finally, since changes in white matter properties could theoretically follow a nonlinear trajectory, we tested a model that included a quadratic term for each tract and parameter. For MD in each tract, the linear model outperformed the nonlinear model (evaluated using Bayesian Information Criteria (BIC)[50,51]), and no significant nonlinear effects were observed: AF linear: $F(1,76) = 8.72$, $p = 0.0041$, AF quadratic: $F(1,76) = 0.31$, $p = 0.58$, ILF linear: $F(1,76) = 7.53$, $p = 0.0076$, ILF quadratic: $F(1,76) = 0.33$, $p = 0.57$, CC linear: $F(1,76) = 3.083$, $p = 0.083$, CC quadratic: $F(1,76) = 3.90$, $p = 0.052$. In contrast, we observed significant quadratic effects in FA for the left AF only: AF linear: $F(1,76) = 3.87$, $p = 0.053$, AF quadratic: $F(1,76) = 7.77$, $p = 0.0067$, ILF linear: $F(1,76) = 8.85$, $p = 0.0039$, ILF quadratic: $F(1,76) = 3.20$, $p = 0.078$, CC linear: $F(1,76) = 0.31$, $p = 0.58$, CC quadratic: $F(1,76) = 2.047$, $p = 0.16$.

Like the reading outcomes reported above, intervention-driven changes in MD were specific to the intervention group, as indicated by a significant group (intervention vs. control) by time

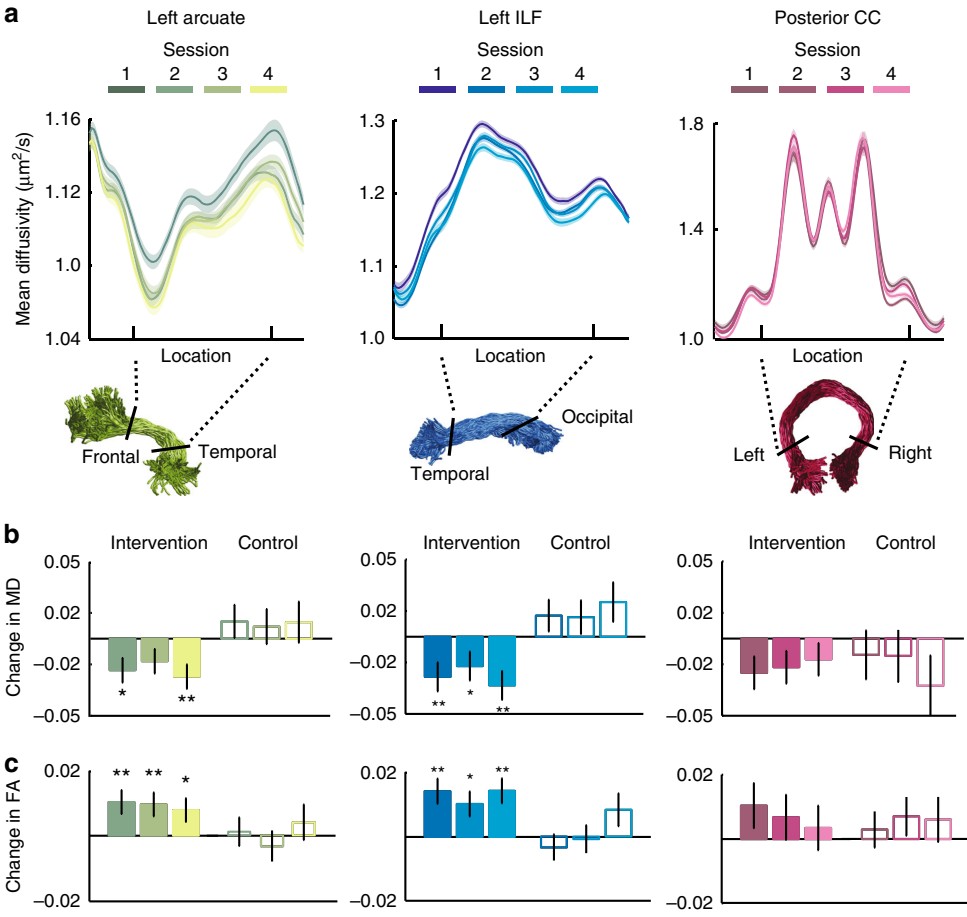

**Fig. 2** Change vs. stability in Tract Profiles during reading intervention. **a** Mean diffusivity (MD) values were mapped onto each of the 100 evenly spaced nodes spanning termination points at the gray–white matter boundary to create a 'Tract Profile' (see Methods and ref. [78] for additional details of this analysis). For visualization purposes, the middle 80 nodes are plotted. Each line represents the group average MD across subjects, measured at four time points: pre-intervention (Session 1), after ~2.5 weeks of intervention (Session 2), after ~5 weeks of intervention (Session 3), and after 8 weeks of intervention (Session 4). Shaded error bars give ±1 standard error of the mean. Color values indicate session, ranging from darkest (Session 1) to brightest (Session 4) for each tract. The x axis shows the location where each tract was clipped prior to analysis (corresponding to black boundary lines in renderings, below). Tract renderings are shown for an example subject. The middle 60% (bounded by black lines) of each tract was analyzed in **b**, **c**, to avoid partial volume effects that occur at endpoints of the tract, where it enters cortex. Both the arcuate fasciculus (AF) and inferior longitudinal fasciculus (ILF), but not the posterior callosal connections, show a systematic decrease in MD over the course of intervention. **b**, **c** Bars show model-predicted change (coefficients and standard errors from mixed effects model) in MD (**b**) and FA (**c**) for each session. Bar heights represent the magnitude of change observed in that session, relative to Session 1 (pre-intervention) baseline, as determined by the mixed effects model. As described in the main text, both the AF and ILF showed significant change between sessions for the intervention group (filled bars) but not the control group (unfilled bars). Asterisks indicate a significant decrease in MD (**b**) or increase in FA (**c**) for each session relative to the pre-intervention baseline at a Bonferroni-corrected *$p < 0.05$ and **$p < 0.01$

(days) interaction. As above, we substitute "days" for "intervention hours" to give a meaningful predictor for both the intervention and control subjects. In the left AF, we found a significant main effect of group ($F(1,125) = 7.047$, $p = 0.009$) but not of time ($F(1,125) = 1.033$, $p = 0.31$) and a significant group-by-time interaction ($F(1,125) = 4.97$, $p = 0.028$), consistent with a decrease in MD over time that was specific to the intervention subjects. Similarly, in the ILF, we saw a significant main effect of group ($F(1,125) = 10.29$, $p = 0.0017$) but not of time ($F(1,125) = 3.72$, $p = 0.056$) and a significant group-by-time interaction ($F(1,125) = 9.53$, $p = 0.0025$). In the CC, we saw a significant main effect of group ($F(1,125) = 6.69$, $p = 0.011$) but not of time ($F(1,125) = 0.90$ $p = 0.34$) and no significant group-by-time interaction ($F(1,125) = 0.027$, $p = 0.87$), consistent with the stability of MD values in this tract in all subjects. For FA, we observed a different pattern of results: In the AF, we saw no significant main effect of group ($F(1,125) = 0.31$, $p = 0.58$) or time ($F(1,125) = 0.055$, $p = 0.82$) and no significant group-by-time interaction ($F$

$(1,125) = 0.36$, $p = 0.55$). In the ILF, we saw no significant main effect of group ($F(1,125) = 0.0015$, $p = 0.97$) or time ($F(1,125) = 1.93$, $p = 0.17$) and no significant group-by-time interaction ($F(1,125) = 0.15$, $p = 0.70$). In the CC, we saw no significant main effect of group ($F(1,125) = 0.23$, $p = 0.63$) or time ($F(1,125) = 0.86$, $p = 0.36$) and no significant group-by-time interaction ($F(1,125) = 0.35$, $p = 0.56$). As shown in Supplementary Table 3, the group-by-time interaction approached significance for the quadratic term for FA in the left AF and ILF, but not for MD in the AF or ILF, or for either parameter in the CC.

Given the observed non-linearity of intervention-driven effects in FA, we opted to use "session number" as a categorical predictor in the analysis to follow, since this approach summarizes session-to-session differences from baseline, without imposing a shape on the trajectory of change. Sessions were systematically spaced over time, and this timing was consistent across subjects; hence "session" was highly correlated with "days" ($r(127) = 0.97$, $p < 0.001$). As shown in Fig. 2, both the left AF and ILF showed

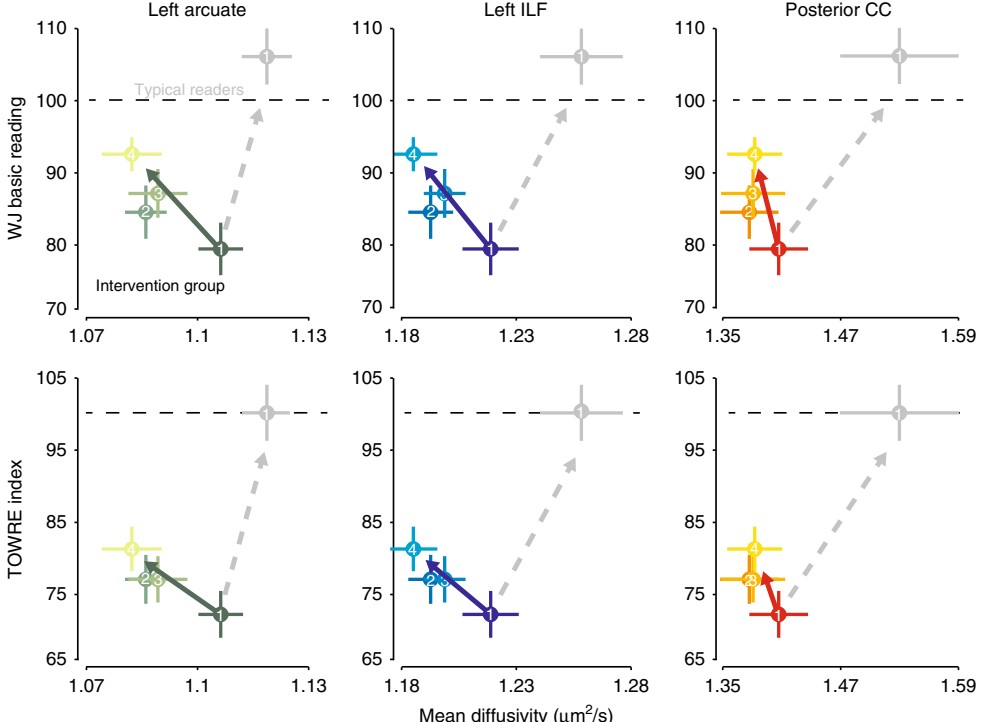

**Fig. 3** Reading intervention does not normalize differences in the white matter. Reading skill is plotted as a function of mean diffusivity for each session (1–4) for the left arcuate, ILF, and posterior callosal tracts in the intervention group. The gray circle and lines in each panel show the mean and standard error at Session 1 for the subset non-intervention control subjects with reading skills in the typical range (poor reading controls were excluded, leaving $n = 10$ typical reading controls). The dashed gray arrow shows the expected trajectory for MD values if the intervention group were to become more similar to the typical reading controls in terms of both reading skills and MD. In contrast, the true trajectory of change in plotted as a colored arrow in each panel. The intervention group includes some readers with only moderate reading impairments (and, therefore, higher MD values), and so the group difference in pre-intervention scores is less than would be expected for a group of good vs. poor readers. Supplementary Fig. 7 reproduces the intervention data alongside the same mean and SE plots for the full sample of control subjects during sessions 1–3, to further illustrate the relative stability of both white matter and behavior in the control group

clear intervention-driven changes in both MD and FA. Within the intervention group, significant changes in tissue properties emerged in the first post-baseline measurement session, after just 46.05 h (SD = 14.88) of intervention, over the course of 2–3 weeks. In line with the results reported above for the continuous predictor (days), we observed a group-by-session interaction for MD in the AF (no main effect of session, $F(1,67) = 2.12$, $p = 0.15$, or group, $F(1,67) = 0.58$, $p = 0.45$, session-by-group interaction, $F(1,67) = 7.75$, $p = 0.0070$) and the ILF (no main effect of session, $F(1,67) = 1.77$, $p = 0.19$, or group, $F(1,67) = 0.044$, $p = 0.83$, session-by-group interaction, $F(1,67) = 6.91$, $p = 0.011$) but not the CC (no main effect of session, $F(1,67) = 1.029$, $p = 0.31$, main effect of group, $F(1,67) = 5.99$, $p = 0.017$, no session-by-group interaction, $F(1,67) = 0.62$, $p = 0.44$) and for FA in the ILF (no main effect of session, $F(1,67) = 0.65$, $p = 0.42$, or group, $F(1,67) = 0.60$, $p = 0.44$, session-by-group interaction, $F(1,67) = 6.45$, $p = 0.013$) but not the AF (no main effect of session, $F(1,67) = 0.0057$, $p = 0.94$, or group, $F(1,67) = 1.57$, $p = 0.21$, no session-by-group interaction, $F(1,67) = 2.85$, $p = 0.096$) or CC (no main effect of session, $F(1,67) = 0.26$, $p = 0.61$, or group, $F(1,67) = 0.14$, $p = 0.71$, no session-by-group interaction, $F(1,67) = 2.38$, $p = 0.13$). An exploratory analysis of this same session-by-group interaction for all available tracts is given in Supplementary Table 10. Finally, to ensure that the interaction was not driven by differences in the stability of our measurements in good vs. poor readers, given that the control group included both typical readers and subjects with dyslexia, we repeated the above analysis with baseline Reading

Skill included as a covariate in the model. We obtained the same results for the group-by-session interaction in all cases (AF: MD, $F(1,65) = 7.72$, $p = 0.0071$; FA, $F(1,65) = 2.86$, $p = 0.095$; ILF: MD, $F(1,65) = 8.37$, $p = 0.0052$; FA, $F(1,65) = 6.71$, $p = 0.012$; CC: MD, $F(1,65) = 0.63$, $p = 0.43$; FA, $F(1,65) = 2.42$, $p = 0.12$).

**Relationship between white matter plasticity and remediation.** One possible interpretation of group differences in MD and FA between good and poor readers is that these differences reflect abnormal tissue properties in poor readers. In that case, one might predict that remediation of reading difficulties would involve a "normalization" of deficits in white matter structure. Alternatively, plasticity in the white matter might reflect a compensatory mechanism that supports the learning process[12,52,53]. In that case, white matter tissue properties in the remediated readers would not necessarily look more similar to those in the typical reading control subjects.

We find that intervention-driven changes in white matter deviate from the trajectory predicted by a normalization account. Figure 3 shows changes in MD and reading scores for the intervention group, relative to Session 1 data for the subset of non-intervention controls who had reading skills in the typical range. We defined "Typical Readers" as Control Group subjects with timed (TOWRE Index) and untimed (WJ Basic Reading Score) reading accuracy within a standard deviation of the population mean (at or above 85 on both measures). For the intervention group, we plot changes in both WJ Basic Reading

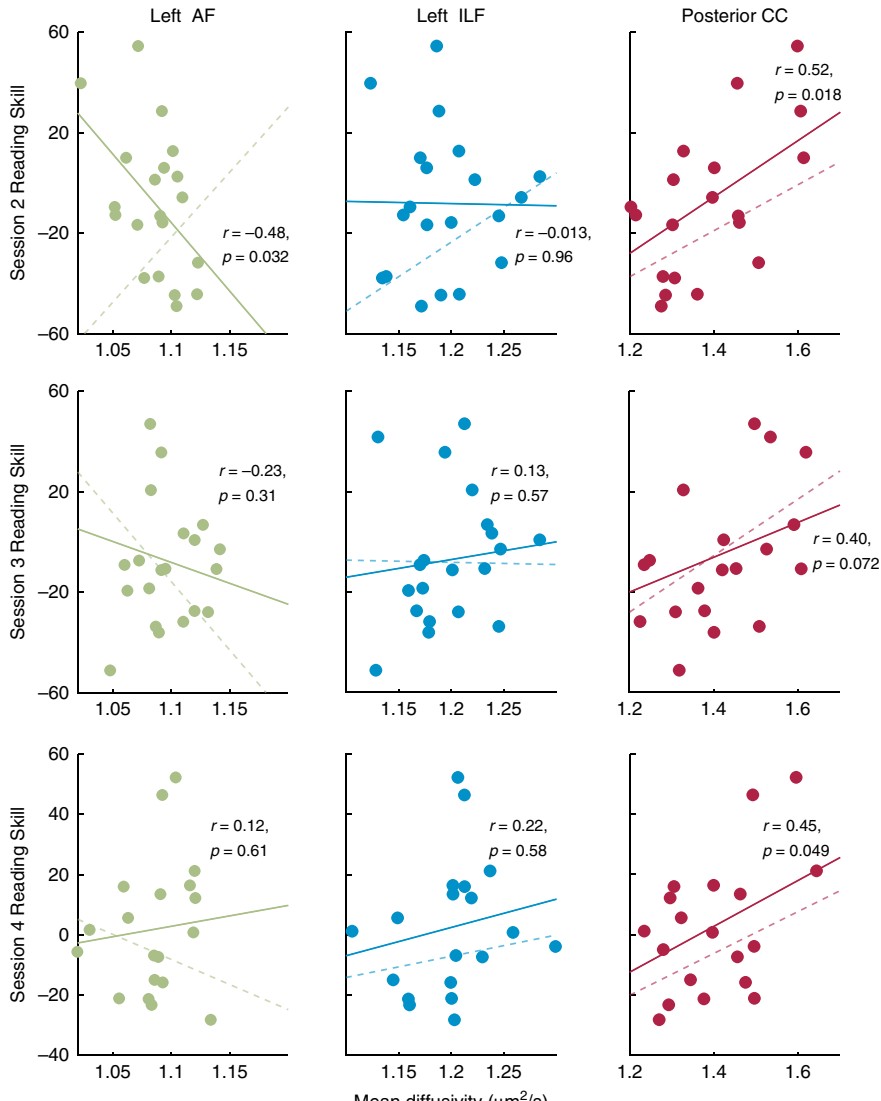

**Fig. 4** Correlations between white matter properties and reading skill change during learning. Plots show the cross-sectional correlation for sessions 2–4 (top to bottom) for each tract for the intervention group. A solid best-fit line is plotted for each session. A dashed line in each panel represents the best-fit line for the preceding session (session *n*−1) to illustrate the session-to-session changes. For the AF and ILF, correlations change in size and/or direction, demonstrating that anatomy–behavior relationships can depend on recent educational experience

and the TOWRE Index (rather than composite Reading Skill) in order to situate the cross-sectional and intervention-driven effects relative to an age-normed, population mean. After completing the intervention, tissue properties in the intervention subjects were not more similar to the typical reading controls, despite a substantial improvement in reading skills. As diffusion properties such as MD are influenced by multiple biological sources, this finding indicates that short-term plasticity is likely to reflect a different biological mechanism than the group differences reported here and in other studies. Further, the short-term, experience-dependent changes in the white matter were larger (Cohen's $d = 0.75$ for the AF, and $d = 0.66$ for the ILF) than the typical group difference reported in the literature[4,13,46] and the group differences observed here ($d = 0.53$ for the AF and $d = 0.59$ for the ILF). These results demonstrate that the effects of recent experience on measured tissue properties may equal or exceed effects due to intrinsic or long-term maturational factors, suggesting that group differences measured in cross-sectional studies may, in some cases, be driven by systematic differences in environmental influences between groups.

**Anatomy–behavior correlations depend on recent experience.** Over the course of the intervention, only the posterior CC retained a relationship to Reading Skill. In contrast, as MD values declined in the AF and ILF, the instantaneous, cross-sectional correlation between reading and MD changed between sessions, as indicated by a significant interaction between MD and session in predicting Reading Skill for the intervention group (linear mixed effects model predicting Reading Skill from MD, session, and their interaction, with a random effect of subject, see Fig. 4). For MD in both the AF and ILF, but not the CC, this interaction was significant (main effect of MD in AF, $F_{(1,71)} = 4.59$, $p = 0.036$, main effect of session $F_{(3,71)} = 28.048$, $p < 10^{-11}$, session-by-MD interaction, $F_{(3,71)} = 2.95$, $p = 0.039$; main effect of MD in ILF, $F_{(1,71)} = 3.97$, $p = 0.050$, main effect of session $F_{(3,71)} = 28.53$, $p < 10^{-11}$, session-by-MD interaction, $F_{(3,71)} = 3.56$, $p = 0.018$; main effect of MD in CC, $F_{(1,71)} = 1.56$, $p = 0.22$, main effect of session $F_{(3,71)} = 27.19$, $p < 10^{-11}$, session-by-MD interaction, $F_{(3,71)} = 0.64$, $p = 0.59$). In the AF, this effect was also significant for FA (main effect of FA in AF, $F_{(1,71)} = 8.48$, $p = 0.0047$, main effect of session $F_{(3,71)} = 31.91$, $p < 10^{-13}$,

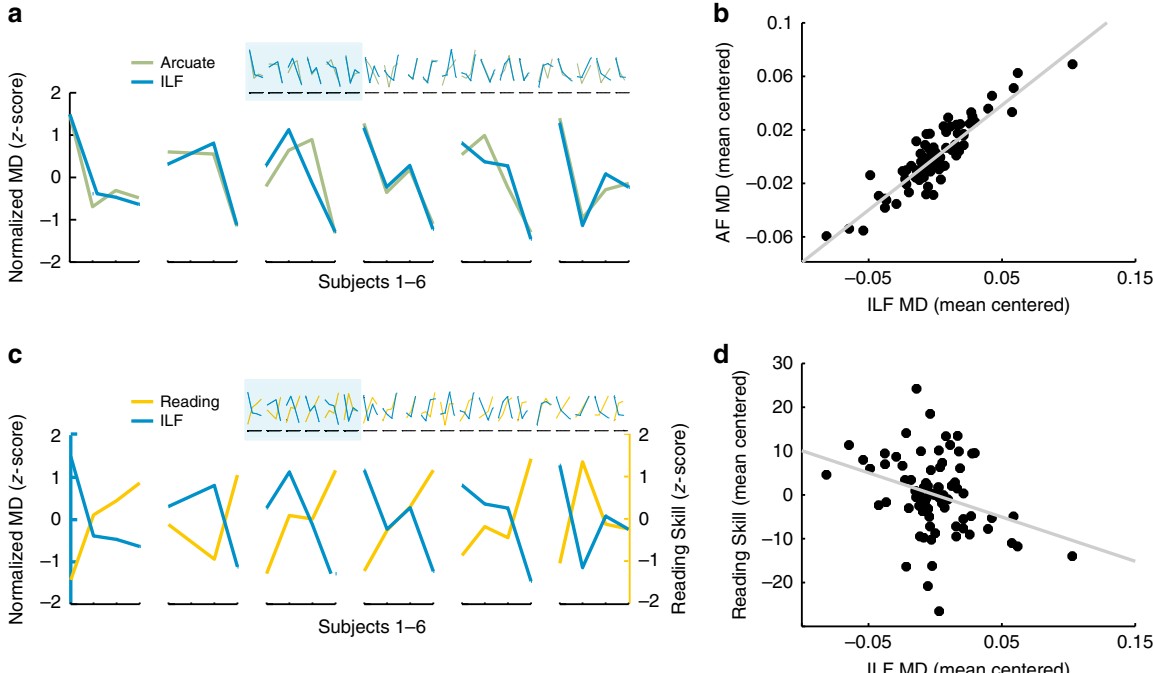

**Fig. 5** Analysis of individual growth trajectories in the AF and ILF. **a** Left arcuate and ILF mean diffusivity (MD) time courses are correlated across individual subjects. For visualization, standardized MD values are plotted for each tract at each available time point for all subjects. Individual time courses are enlarged for a set of six example subjects to show greater detail. Individual AF and ILF time courses were positively correlated, and a cross-correlation analysis failed to detect a significant correlation at any non-zero lag, consistent with the interpretation that growth occurs in concert across tracts. **b** MD values were mean-centered for each subject, thus representing each subject's time course of MD changes as modulations around their mean. Mean-centered values at each available time point are plotted for the AF vs. ILF showing tight correspondence between changes in these two tracts. **c** The time course of change in Reading Skill varies across subjects and is negatively related to individual white matter time courses: Within a subject, as MD decreases in the ILF, Reading Skill increase. For visualization, standardized Reading Skill and MD values are plotted for the ILF at each available time point for all subjects. As in **a**, time courses are enlarged for a set of six example subjects to show individual subject trajectories in greater detail. **d** Mean-centered MD values at each available time point for the ILF correlate with mean-centered Reading Skill assessed at each time point, demonstrating the relationship between time courses of MD and Reading Skill change. The scatter plots in **b**, **d** also make it clear that the time course of plasticity is more tightly coupled across tracts than it is to behavior. Hence, even though there is a statistically significant relationship between the time course of white matter and behavioral changes, there is also un-explained variance that is likely to be related to aspects of the intervention environment that do not directly impact behavior

session-by-FA interaction, $F(3,71) = 4.28$, $p = 0.0078$; main effect of FA in ILF, $F(1,71) = 7.44$, $p = 0.0080$, main effect of session $F(3,71) = 30.99$, $p < 10^{-13}$, session-by-FA interaction, $F(3,73) = 1.81$, $p = 0.15$; main effect of FA in CC, $F(1,71) = 9.43$, $p = 0.003$, main effect of session $F(3,71) = 32.60$, $p < 10^{-12}$, session-by-MD interaction, $F(3,71) = 2.077$, $p = 0.11$). Importantly, changes in both the strength and the sign (positive vs. negative) of observed correlations could not be attributed simply to session-by-session changes in variance of reading skills or white matter. As shown in Supplementary Table 4, there was no statistical difference in variance across sessions (indeed, variances were nearly matched; see also Fig. 3, which plots means and errors for each session). Therefore, changing anatomy–behavior correlations were not driven by differences in relative variance over time, and instead reflect learning-related dynamics in the reading and white matter measures.

Finally, we found no evidence for changing anatomy–behavior correlations in the group of children who were not enrolled in the intervention (AF: MD: $F(3,41) = 0.75$, $p = 0.53$, FA: $F(3,41) = 0.12$, $p = 0.95$; ILF: MD: $F(3,41) = 1.36$, $p = 0.27$, FA: $F(3,41) = 1.70$, $p = 0.18$; CC: MD: $F(3,41) = 0.55$, $p = 0.65$, FA: $F(3,41) = 0.97$, $p = 0.42$). This is consistent with the stability of diffusion properties in this group and supports the notion that the significant interaction for the intervention subjects did not arise due to differences in measurement noise over time. Finally, to rule out the possibility that systematic differences in head motion

might influence anatomy–behavior correlations (e.g., children with lower reading scores might move more in the scanner than children with higher reading scores), we calculated the correlation between head motion and Reading Skill. Motion and Reading Skill were unrelated ($r(97) = 0.13$, $p = 0.19$).

**Widespread changes in the white matter track learning**. The AF and ILF connect distinct components of the reading circuitry and are thought to carry signals that contribute uniquely to the reading process[27,40,48]. Therefore, a reading intervention might affect these tracts differently, prompting changes that reflect independent biological processes unfolding with different time courses and reflecting different aspects of learning. To address this possibility, we asked whether changes in the AF and ILF occur in synchrony in the intervention group. If wholly independent mechanisms were driving growth in both tracts, we would not expect to see similar time courses of growth for the AF and ILF within subjects. Alternatively, if changes within the AF and ILF reflect a common biological mechanism operating over a large anatomical scale, then time courses of growth should be correlated within subjects.

To address these questions, we fit a linear mixed effects model to all intervention subjects' mean-centered diffusion measurements over all time points. This allowed us to quantify the similarity between AF and ILF longitudinal growth trajectories while excluding between-subject differences in baseline diffusion properties[54]. Results for a complementary analysis, examining

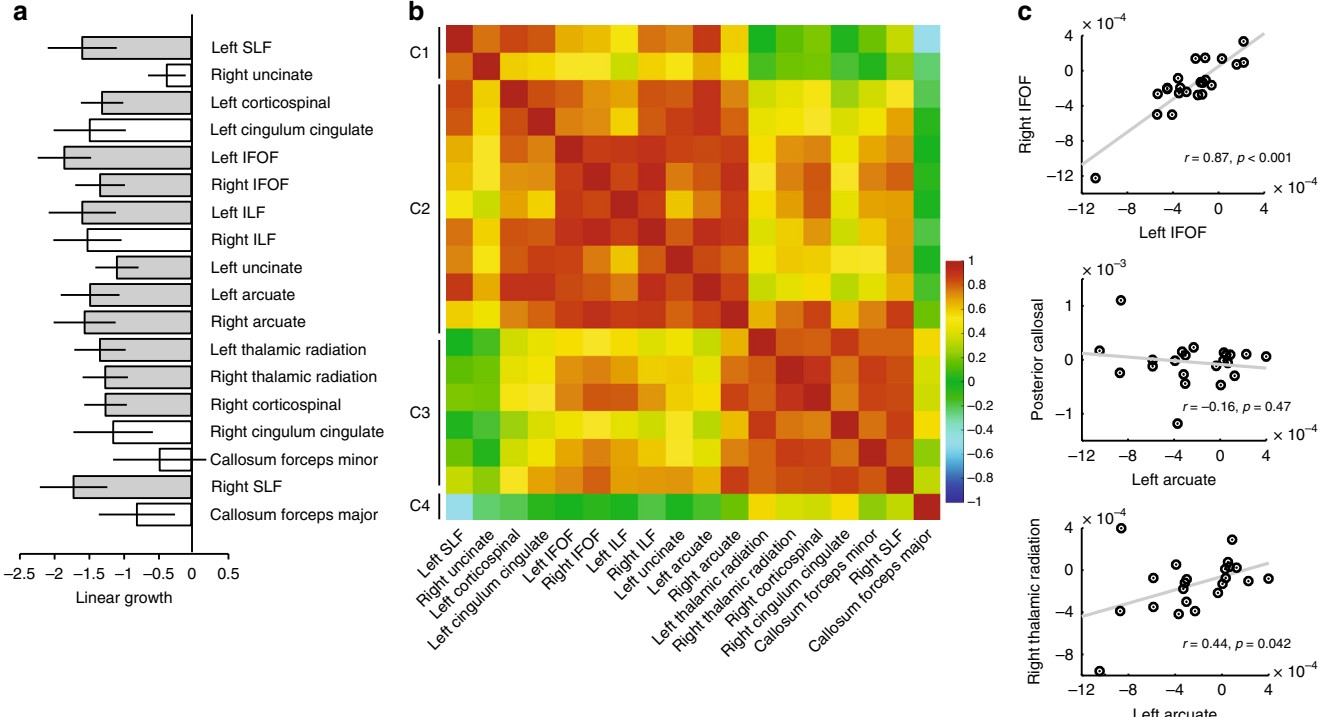

**Fig. 6** Reading intervention causes distributed changes in the white matter. **a** Change in MD as a function of intervention time (in hours) for 18 tracts. Tracts showing significant change (Bonferroni-corrected $p < 0.05$) are indicated as gray filled bars. Error bars depict standard errors from a linear mixed effects model. **b** Hierarchical clustering based on the correlations between linear growth rates. The heat map represents Pearson correlations between linear growth rates for pairs of tracts across individuals. The matrix is sorted according to hierarchical clustering of these correlation coefficients. **c** Scatter plots of individual growth rates for three pairs of tracts: left vs. right IFOF, AF vs. CC, and AF vs. right thalamic radiation

diffusion measurements relative to a pre-intervention baseline, are given in Supplementary Table 5. As shown in Fig. 5, the time courses of change in the AF and ILF were highly correlated for both MD and FA (MD: $r = 0.86$, $p < 0.001$; FA: $r = 0.50$, $p = 0.021$), implying that, within each individual, white matter growth trajectories were tightly coupled for these two tracts. We then fit the same model for time-lagged versions of each tract's time course to test whether these regions changed in synchrony. If growth in one tract were to precede growth in the other, this would imply a distinct and more gradual process occurring in the second tract or a possible causal relationship. In that case, the time courses should be better predicted by time-lagged versions of each other. However, we failed to detect a significant correlation at any non-zero lag, suggesting that these tracts change in concert as a function of experience in the reading intervention program.

Finally, to examine the relative timing of white matter changes in relation to learning, we performed the same cross-correlation analysis with the reading scores: Each intervention subject's reading scores were mean-centered to remove inter-subject differences in baseline reading ability, and a linear mixed effects model was fit to shifted (lag = −1 and lag = 1) and un-shifted (lag = 0) versions of the time courses. Time courses of MD, but not of FA, were significantly correlated with time courses of Reading Skill only at lag = 0 (MD: $r = -0.30$ Arcuate, $p = 0.0069$; $r = -0.30$ ILF, $p = 0.0061$), demonstrating that, within a subject, the time course of white matter plasticity tracked the time course of learning. For MD, we again found that the growth trajectories were best fit by the un-shifted time courses, suggesting that white matter changes are coupled to reading experience and, therefore, track improvements in Reading Skill. In the control group, no tracts showed a significant relationship to reading skill at any lag (shown for lag = 0 in Supplementary Table 6), consistent with the

stability of both reading and white matter properties in control subjects.

Since a substantial proportion of the total changes in MD occurred during the first 2 weeks of intervention, we also examined the relationship between reading and white matter changes during this interval by correlating Session 2 vs. Session 1 difference scores. Individual differences in the magnitude of Sessions 2–1 MD change were not significantly correlated with the magnitude of reading score change. As shown in Supplementary Table 7, we observed a trend for both raw and standardized reading scores. Since this analysis only includes half of the data, we cannot ascertain whether the result represents the absence of a relationship at this short timescale or the lack of statistical power.

**Trajectories of plasticity are correlated across tracts**. White matter growth rates were highly correlated for two tracts considered to be critical for skilled reading, meaning that a subject showing rapid, intervention-driven growth in the AF also shows considerable growth in the ILF. However, changes in MD and FA were not limited to the connections of the core reading circuitry; instead, we observed significant change throughout a collection of tracts, extending beyond our a priori hypothesis. Figure 6a models growth in MD as a linear function of the number of intervention hours, and we use a conservative Bonferroni correction in this exploratory analysis. In the intervention group, 12 out of the 18 tracts showed significant (Bonferroni corrected) change. None of the 18 tracts showed significant change in either MD or FA in the control group. Further, as shown in Table 1, multiple tracts showed a significant relationship to changes in reading skill, including, but not limited

**Table 1 White matter properties track changes in Reading Skill**

| Tract (MD) | Reading Skill | WJ-BRS | TOWRE Index | WJ-RF |
|---|---|---|---|---|
| Left thalamic radiation | $r = -0.025, p = 0.026$ | $r = -0.16, p = 0.17$ | $r = -0.38, p = 0.00056$ | $r = -0.15, p = 0.18$ |
| Right thalamic radiation | $r = -0.31, p = 0.0047$ | $r = -0.26, p = 0.02$ | $r = -0.37, p = 0.00067$ | $r = -0.15, p = 0.18$ |
| Left corticospinal | $r = -0.39, p = 0.0003$ | $r = -0.32, p = 0.0040$ | $r = -0.39, p = 0.00028$ | $r = -0.26, p = 0.022$ |
| Right corticospinal | $r = -0.34, p = 0.0019$ | $r = -0.27, p = 0.014$ | $r = -0.40, p = 0.00025$ | $r = -0.21, p = 0.062$ |
| Left cingulum | $r = -0.19, p = 0.083$ | $r = -0.088, p = 0.44$ | $r = -0.34, p = 0.0023$ | $r = -0.11, p = 0.31$ |
| Right cingulum | $r = -0.12, p = 0.29$ | $r = -0.13, p = 0.23$ | $r = -0.11, p = 0.31$ | $r = -0.050, p = 0.66$ |
| Posterior callosal | $r = -0.23, p = 0.040$ | $r = -0.19, p = 0.087$ | $r = -0.21, p = 0.066$ | $r = -0.18, p = 0.10$ |
| Anterior callosal | $r = 0.041, p = 0.72$ | $r = 0.086, p = 0.44$ | $r = -0.092, p = 0.41$ | $r = 0.077, p = 0.50$ |
| Left IFOF | $r = -0.33, p = 0.0024$ | $r = -0.24, p = 0.035$ | $r = -0.42, p = 0.00011$ | $r = -0.14, p = 0.22$ |
| Right IFOF | $r = -0.28, p = 0.013$ | $r = -0.25, p = 0.024$ | $r = -0.25, p = 0.024$ | $r = -0.089, p = 0.43$ |
| Left ILF | $r = -0.30, p = 0.0061$ | $r = -0.19, p = 0.087$ | $r = -0.40, p = 0.00021$ | $r = -0.10, p = 0.37$ |
| Right ILF | $r = -0.26, p = 0.019$ | $r = -0.21, p = 0.058$ | $r = -0.28, p = 0.012$ | $r = -0.054, p = 0.63$ |
| Left SLF | $r = -0.25, p = 0.026$ | $r = -0.19, p = 0.093$ | $r = -0.32, p = 0.0037$ | $r = -0.089, p = 0.43$ |
| Right SLF | $r = -0.25, p = 0.022$ | $r = -0.20, p = 0.077$ | $r = -0.31, p = 0.0047$ | $r = -0.13, p = 0.26$ |
| Left uncinate | $r = -0.29, p = 0.0081$ | $r = -0.24, p = 0.029$ | $r = -0.31, p = 0.0044$ | $r = -0.16, p = 0.17$ |
| Right uncinate | $r = 0.037, p = 0.74$ | $r = 0.066, p = 0.56$ | $r = 0.0051, p = 0.96$ | $r = 0.17, p = 0.13$ |
| Left arcuate | $r = -0.30, p = 0.0069$ | $r = -0.21, p = 0.064$ | $r = -0.40, p = 0.00022$ | $r = -0.14, p = 0.22$ |
| Right arcuate | $r = -0.29, p = 0.0082$ | $r = -0.24, p = 0.030$ | $r = -0.35, p = 0.0015$ | $r = -0.15, p = 0.18$ |

Cells show $p$-values based on a mixed linear model predicting session-to-session changes in the Reading Skill composite, Woodcock–Johnson Basic Reading (WJ-BRS), TOWRE index, and Woodcock–Johnson Reading Fluency (WJ-RF) from changes in mean diffusivity (MD) at each time point during the intervention. Pearson correlations between mean-centered MD and mean-centered reading score are provided as an index of effect size. Tracts that predict changes in readings scores at a Bonferroni-corrected $p < 0.05$ are highlighted in bold italic

to, the core circuitry for reading. (See Supplementary Table 8 for a complementary analysis relating FA and Reading Skill). Therefore, learning effects are not specific to tracts that are considered to be the core circuitry for reading, and intervention-driven changes are evident in an extensive collection of white matter tracts.

Given that intervention effects appear to be spatially widespread and that changes within two key tracts, the AF and ILF, are tightly coupled, we next examined the correlation structure across the full collection of tracts showing intervention-driven growth. Specifically, we tested whether growth rates are solely coupled within the AF and ILF, versus a larger collection of tracts. To that end, we fit linear growth rates (change in MD or FA as a function of hours of intervention) to each subject's data for the 18 tracts and then computed the correlation between growth rates across each pair of tracts. To assess the suitability of a linear model, we used BIC[50,51] to evaluate the linear model relative to two non-linear models, one with a quadratic and one with an additional cubic component. In all tracts with significant intervention-driven effects, the linear model outperformed both the quadratic and cubic models.

Figure 6b shows the correlation between linear growth rates of pairs of tracts across individuals. The ordering of the tracts was determined according to a hierarchical clustering of these correlation coefficients. This analysis revealed that many tracts show highly correlated intervention-driven changes ($r > 0.7$) and identified a cluster containing many of the cortical association tracts (the left and right ILF, superior longitudinal fasciculus (SLF), IFOF, and arcuate, as well as the left uncinate and left corticospinal tracts), which all changed in concert. In addition, we identified a separate cluster of tracts whose properties change during the intervention, but with independent growth rates. For example, highly significant growth rates are observed bilaterally in the thalamic radiation, but these growth rates are not correlated with growth measured in the left arcuate (Fig. 6c). Accordingly, these tracts are assigned to distinct clusters. We suggest that changes within these distinct clusters may reflect distinct biological mechanisms. A complementary analysis of FA is provided in Supplementary Fig. 2 and identifies a consistent clustering of the tracts.

## Discussion

Intensive reading training causes rapid changes in tissue properties within the left AF and ILF, two tracts considered crucial for skilled reading. However, the effects of intervention are not limited to these regions. Instead, we find widespread change throughout multiple cortical association and projection tracts. Importantly, within individuals, intervention-driven effects are tightly coupled across this collection of tracts. Further, tissue properties and reading skills change in concert: An individual's time course of white matter changes tracks their time course of changes in reading skill. This suggests that the white matter rapidly adapts to the changing environmental demands posed by the intervention. The extent of plasticity in the white matter has important implications for the interpretation of correlations between white matter tissue properties and academic skills: As cross-sectional correlations change week to week, correlations measured at any single time point offer an incomplete, and potentially misleading, view of the underlying relationships between anatomy, behavior, and experience.

Intervention leads to rapid changes that are distributed across cortical association and projection tracts, including, but not limited to, the left AF and left ILF. These tracts connect distinct components of the reading circuitry and are generally considered to support separable aspects of reading. For example, the AF has been linked specifically to phonological awareness[4,26], while the ILF, which projects to the VWFA[40], may be especially involved in visual word recognition. Typically, over years of development, growth rates for these two tracts are independent from each other[28]. We therefore hypothesized that the learning process might differentially affect tissue properties within these tracts. Further, given the diversity of behavioral profiles seen in people with dyslexia, subjects could show differing spatial profiles of change. For example, a subject with strong intervention-driven effects within the AF might show smaller effects within the ILF, while another subject might show the opposite pattern. However, our results support an alternative view. Longitudinal changes in the AF and ILF are tightly coupled within subjects and also correlated to changes in many other white matter tracts, suggesting that these effects arise from a common biological mechanism operating over a large anatomical scale.

Typically, dMRI studies of the white matter seek to identify a single critical structure that is related to a specific cognitive skill. Our measurements offer a different view on white matter plasticity and learning: Anatomically widespread effects may be a hallmark of rapid, short-term plasticity associated with intensive training of reading skills. Since reading depends on the coordination of a large cortical network, training of reading skills may prompt particularly widespread effects across the white matter. Functional changes measured with fMRI after reading training appear to be widespread[55], affecting multiple sites within the cortical and subcortical reading network. However, a relatively small and focal change in anatomy could theoretically produce widespread functional changes, and therefore these effects need not be accompanied by large-scale anatomical remodeling. Indeed, a small number of past studies in human subjects have reported focal changes in white matter after training of reading skills[23,56], but past work has not employed the intensive training paradigm used here (see also ref. [24]). Alternatively, the widespread effects may reflect general mechanisms of learning during an intensive educational experience and therefore may not be specific to the curriculum of this reading intervention.

It is important to note that the tracts identified in this analysis, including the left hemisphere ILF, SLF, and AF, carry signals that are relevant for a number of cognitive functions[57], not only reading[58–60]. Interestingly, individual differences in plasticity within the left AF have recently been linked to individual gains in math skills following math intervention[61], even though the left AF is conventionally associated with language-related skills. It should be noted, however, that in ref. [61], math skills' training did not produce a significant change in the arcuate at a group level, and therefore the previous set of findings differ from ours. Given the relatively coarse (mm) scale of dMRI, it is possible that distinct types of intervention (e.g., training in reading vs. math skills) affect distinct subpopulations of fibers with distinct cortical terminations and functional roles. However, an alternative interpretation also emerges from the current study: Intensive training may lead to plasticity within regions that are not necessarily critical for performing the trained task, and thus intervention-driven effects in the left AF might reflect general mechanisms that are common to learning both reading and math. Despite the lack of a group-level intervention effect in the left arcuate in ref. [61], it remains possible that a sufficiently intense math intervention might prompt changes not only within the left arcuate but also within many of the same tracts identified here. Indeed, our effects may reflect the intensity and quality of the learning environment, rather than the specific trained skills. Moreover, since it would not be feasible to enroll skilled readers in a highly intensive reading intervention program, it is unclear whether the observed effects are specific to individuals with reading difficulties. Future work examining the generalizability of these effects in other domains, such as math, would allow an examination of general learning effects in a broader population and should help clarify the role of domain-specific deficits.

What biological mechanism might underlie the observed effects? Changes in the diffusion signal can arise from multiple sources, including use-related swelling and branching of glial cells[17,18,21,63], changes in vasculature, myelination of unmyelinated axons, myelin remodeling, and/or growth of new myelinating oligodendrocytes (reviewed in ref. [63]). Oligodendrocyte precursor cells are present throughout the white matter, and large-scale proliferation of oligodendrocytes has been shown in mice within hours of optogenetically stimulated activity in adult motor cortex[16]. Mature oligodendrocytes, in turn, participate in myelin maintenance and remodeling throughout the life span. Thus a particularly intriguing possibility is that an initial pattern of widespread change in diffusion properties reported here might reflect rapid growth of myelinating oligodendrocytes, of which only a fraction will ultimately contribute to new myelin sheaths within focal, task-relevant regions. In that case, it should be possible to differentiate diffusion signal changes related to rapid growth of oligodendrocytes from signal changes related to longer-term changes in myelin after the training period has ended. In particular, we might expect a rapid initial large decrease in MD, since diffusion would be hindered by new oligodendrocytes. Subsequent changes in myelin might emerge as relatively smaller, persistent changes in other quantitative MRI[64,65,66].

Within the intervention group, changes in the white matter are linked to changes in behavior (Fig. 5). However, these effects do not follow the trajectory predicted by a normalization account, in which remediation of reading difficulties could be expected to eliminate differences in the white matter between children with dyslexia and typical readers. A number of functional imaging studies have previously examined the extent to which behavioral remediation involves normalization of neural deficits, vs. development of different or compensatory mechanisms[55]. Several studies have reported normalization of responses after successful remediation[53,67], but many studies also highlight neural changes in regions that are not considered fundamental components of the circuit in skilled readers. These learning-induced changes have been interpreted as reflecting compensatory responses[52,53]. Our results are compatible with the idea that remediation may be accomplished through compensatory mechanisms that differ from those supporting the acquisition of skilled reading in typical children.

Given the magnitude of short-term, learning-induced changes in the white matter, previously reported group differences may be driven in part by environmental differences between groups, since systematic difference in the environment (e.g., differences in the quality or intensity of recent educational experiences for dyslexic vs. control subjects) could be expected to exert a large influence on diffusion measurements and to potentially counteract or change pre-existing anatomy–behavior relationships. This offers an explanation for why some studies find a positive correlation between FA and reading skills[10,13], while other studies find a negative correlation between FA and reading skills[4,68] within the exact same tracts. In a single group of subjects, we find that measured anatomy–behavior correlations change over the course of learning. Since learning effects do not "normalize" the white matter, differences in the timing of learning effects among subjects naturally lead to changes in the correlations computed at any given time point. An important implication of the present work is that anatomy–behavior correlations taken at a single time point should be interpreted cautiously, so long as the relevant behaviors and anatomical properties are subject to experience-dependent change.

In contrast to the widespread changes described above, we find that the posterior CC are remarkably stable over the course of intervention and also show stable correlations with reading skills. Although we interpret this null result cautiously, one possibility is that differences in MD within posterior CC reflect relatively stable anatomical variation, which predicts reading skill, but does not change during short-term, intensive training. Indeed, the structure of the posterior corpus callosum differs in both children and adults with dyslexia, and the correlation between diffusion properties in this pathway and reading skills has been reported by many other studies[2,29,69,70]. These connections are known to mature relatively early; therefore, the subjects in our study may already be outside the sensitive period in which experience shapes these connections. In that case, training at an earlier age might prompt changes in the CC alongside acquisition of reading skills.

In summary, our results show that altering a child's educational environment through a targeted intervention program induces

**Table 2 Subject demographics and pre-intervention test scores**

| Subject group | Age in years | WJ-BRS | WJ-RF | TOWRE |
|---|---|---|---|---|
| Intervention | 9.71/1.80 | 79.45/16.27 | 72.05/21.065 | 72.25/14.48 |
| Control | 9.95/1.36 | 95.17/16.87 | 91.17/18.63 | 87.44/18.18 |
| Intervention vs. control | $t(36) = -0.47$, $p = 0.64$ | $t(36) = -2.92$, $p = 0.0060$ | $t(36) = -2.95$, $p = 0.0056$ | $t(36) = -2.86$, $p = 0.0069$ |

Age and pre-intervention Woodcock–Johnson Basic Reading Skills (WJ-BRS), Reading Fluency (WJ-RF), and Test of Word Reading Efficiency (TOWRE) composite standard scores are given for the intervention and control groups. Means are given, followed by sample standard deviation (mean/SD). Intervention and control groups were matched in age, but the intervention group had significantly lower reading scores

rapid, large-scale changes in white matter tissue properties. We observe changes in both MD and FA that occur over the timescale of weeks, that track changes in an individual's reading skills, and that are tightly coupled across tracts connecting distinct parts of the neural circuitry for reading.

## Methods

**Participants.** A total of 93 behavioral and MRI sessions were conducted with a group of 24 children (11 females), ranging in age from 7 to 12 years, who participated in an intensive summer reading intervention program. Members of the intervention group were recruited based on parent report of reading difficulties and/or a clinical diagnosis of dyslexia. An additional 52 behavioral and MRI sessions were conducted with 19 participants, who were matched for age but not reading level. These subjects were recruited as a control group to assess the stability of our measurements over the repeated sessions. Control subjects participated in the same experimental sessions, but did not receive the reading intervention. Ten of these subjects had typical reading skills (4 females), defined as a score of 85 or greater on the Woodcock–Johnson Basic Reading composite and the TOWRE Index. Nine had reading difficulties (3 females), defined as a score below 85 on either the Woodcock–Johnson Basic Reading composite or the TOWRE Index. Reading assessments were carried out at the start of the intervention period to confirm parent reports and establish a baseline for subsequent estimates of growth in reading skill. Demographics and initial test scores are summarized in Table 2.

All participants were native English speakers with normal or corrected-to-normal vision and no history of neurological damage or psychiatric disorder. We obtained written consent from parents and verbal assent from all child participants. All procedures, including recruitment, consent, and testing, followed the guidelines of the University of Washington Human Subjects Division and were reviewed and approved by the UW Institutional Review Board.

**Reading intervention.** Intervention subjects were enrolled in 8 weeks of the Seeing Stars: Symbol Imagery for Fluency, Orthography, Sight Words, and Spelling[71] program at three different Lindamood-Bell Learning Centers in the Seattle area. The intervention program consists of directed, one-on-one training in phonological and orthographic processing skills, lasting 4 h each day, 5 days a week. The curriculum uses an incremental approach, building from letters and syllables to words and connected texts, emphasizing phonological decoding skills as a foundation for spelling and comprehension. A hallmark of this intervention program is the intensity of the training protocol (4 h a day, 5 days a week) and the personalized approach that comes with one-on-one instruction.

**Experimental sessions.** Subjects participated in four experimental sessions separated by roughly 2.5-week intervals. For the intervention group, sessions were scheduled to occur before the intervention (baseline), after 2.5 weeks of intervention, after 5 weeks of intervention, and at the end of the 8-week intervention period. For the control group, sessions followed the same schedule while the subjects attended school as usual. This allowed us to control for changes that would occur due to typical development and learning during the school year. Twenty-one intervention subjects completed all four experimental sessions; 3 subjects completed only 3 sessions, which fell at the start, middle, and end of the intervention. In the control group, 7 subjects completed all 4 sessions; 12 subjects completed at least 3 sessions; 14 subjects completed at least 2 sessions; 19 subjects completed at least one session.

In addition to MRI measurements, described in greater detail below, we administered a battery of behavioral tests in each experimental session. These included subtests from the Wechsler Abbreviated Scales of Intelligence, Comprehensive Test of Phonological Processing (CTOPP-2), TOWRE-2 and the Woodcock–Johnson IV Tests of Achievement (WJ-IV). Rather than analyzing each subtest individually, we created a general reading skills index by conducting a principal component analysis on subtests from the latter two batteries (TOWRE and WJ-IV) and taking scores from the first principal component, which accounted for 83.76% of the total variance in reading performance (Supplementary Fig. 1). We used this measure for all subsequent analysis in order to avoid issues that arise from multiple comparisons and to increase the reliability of our outcome variable.

Our Reading Skill index was highly correlated with both the WJ-BRS composite ($r$ (97) = 0.95, $p < 0.001$) and the TOWRE composite ($r(97) = 0.96$, $p < 0.001$).

**MRI acquisition and processing.** All imaging data were acquired with a 3 T Phillips Achieva scanner (Philips, Eindhoven, Netherlands) at the University of Washington Diagnostic Imaging Sciences Center using a 32-channel head coil. An inflatable cap was used to minimize head motion, and participants were continuously monitored through a closed circuit camera system. Prior to the first MRI session, all subjects completed a session in an MRI simulator, which helped them to practice holding still, with experimenter feedback. This practice session also allowed subjects to experience the noise and confinement of the scanner prior to the actual imaging sessions and to help them feel comfortable and relaxed during data collection.

dMRI data were acquired with isotropic 2.0 mm$^3$ spatial resolution and full brain coverage. Each session consisted of 2 diffusion-weighted imaging (DWI) scans, one with 32 non-collinear directions (b-value = 800 s/mm$^2$) and a second with 64 non-collinear directions (b-value = 2000 s/mm$^2$). The gradient directions were optimized to provide uniform coverage[72]. Each of the DWI scans also contained 4 volumes without diffusion weighting (b-value = 0). In addition, we collected one scan with six non-diffusion-weighted volumes and a reversed phase encoding direction (posterior-anterior) to correct for echo-planar imaging distortions due to inhomogeneities in the magnetic field. Distortion correction was performed using FSL's *topup* tool[73]. Additional pre-processing was carried out using tools in FSL for motion and eddy current correction[74], and diffusion metrics were fit using the diffusion kurtosis model[75] as implemented in DIPY[76]. Data were manually checked for imaging artifacts and excessive dropped volumes. Given that subject motion can be especially problematic for the interpretation of group differences in DWI data[77], datasets with mean slice-by-slice root mean square displacement >0.7 mm were excluded from all further analyses. Datasets in which >10% of volumes were either dropped or contained visible artifact were also excluded from further analysis. In total, these criteria removed 13 out of the 93 total intervention datasets and 3 out of the 52 control datasets.

To further quantify potential effects of motion, we tested for differences in motion across sessions and subject groups (intervention vs. control; see Supplementary Fig. 3), after excluding datasets based on the criteria listed above. We observed no difference in motion as a function of session ($F(3,121) = 0.090$, $p = 0.97$) or group ($F(1,121) = 2.54$, $p = 0.11$) and no group-by-session interaction ($F(3,121) = 0.30$, $p = 0.83$). Thus we do not attribute the between-session changes in white matter within the intervention group to systematic differences in motion. Further, including motion as a covariate in our analysis did not change our results, as described below.

**White matter tract identification.** Fiber tracts were identified for each subject using the Automated Fiber Quantification (AFQ) software package[78], after initial generation of a whole-brain connectome using probabilistic tractography (MRtrix 3.0)[79]. Fiber tracking was carried out on an aligned, distortion corrected, concatenated dataset including all 4 of the 64 direction (b-value = 2000 s/mm$^2$) datasets collected across sessions for each subject. This allowed us to ensure that estimates of diffusivity and diffusion anisotropy across session were mapped to the same anatomical location for each subject, since slight differences in diffusion properties over the course of intervention can influence the region of interest that is identified by the tractography algorithm. We also replicated our main results using tractography derived separately for each session and subject (see Supplementary Fig. 4).

We focused our initial analysis on three tracts that are thought to connect the core reading circuitry[27,28,80]: the left AF, left ILF, and posterior CC. Subsequent analysis included 13 additional tracts: left and right thalamic radiations, left and right corticospinal tracts, anterior CC, left and right IFOF, right ILF, left and right SLF, left and right uncinate, and right AF.

We quantified test–re-test reliability for the full set of 18 tracts in the control group using Pearson's correlation (see Supplementary Table 11). In control subjects, the median reliability across tracts was $r = 0.73$ for MD and $r = 0.76$ for FA.

**Quantifying white matter tissue properties**. To detect intervention-driven changes in the white matter, we fit the diffusion kurtosis model[75] as implemented in DIPY[76] to the diffusion data collected in each session. The diffusion kurtosis model is an extension of the diffusion tensor model that accounts for the non-Gaussian behavior of water in heterogeneous tissue containing multiple barriers to diffusion (cell membranes, myelin sheaths, etc.). After model fitting, diffusion metrics were projected onto the segmented fiber tracts generated by AFQ. Selected tracts were sampled into 100 evenly spaced nodes, spanning termination points at the gray–white matter boundary, and then diffusion properties (mean, radial, and axial diffusivity (MD, RD, AD) and FA) were mapped onto each tract to create a "Tract Profile."

**Statistical analysis**. Data analysis was carried out using software written in MATLAB. To assess change over the course of intervention, we first averaged the middle 60% of each tract to create a single estimate of diffusion properties for each subject and tract. We selected the middle portion to eliminate the influence of crossing fibers near cortical terminations and to avoid potential partial volume effects at the white matter/gray matter border. Mean tract values were then entered into a linear mixed effects model, with fixed effects of intervention time (either hours of training or session entered as a categorical variable) and a random effect of subject. We modeled the relationship between white matter properties and behavior in a similar fashion, predicting Reading Skill from mean tract values and session, with subjects treated as a random effect.

We further examined the time course of change in white matter and reading skills by (1) performing a cross-correlation analysis on individual longitudinal trajectories and (2) calculating individual linear growth rates, which allowed us to directly model relationships between behavioral and white matter growth rates across subjects.

Finally, to examine the anatomical specificity of intervention-driven changes, we fit a mixed linear model to the growth trajectories of a large collection of white matter tracts. We then performed hierarchical clustering on the correlations between linear growth rates, using a complete-linkage clustering algorithm implemented in MATLAB, to test for correlated growth trajectories across a large collection of cortical association tracts.

**Data availability**. All code and data required to reproduce reported findings is available at https://github.com/yeatmanlab/Huber_2018_NatCommun.

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

# ARTICLE

46. Deutsch, G. K. et al. Children's reading performance is correlated with white matter structure measured by diffusion tensor imaging. *Cortex* **41**, 354–363 (2005).

47. Lebel, C. & Beaulieu, C. Longitudinal development of human brain wiring continues from childhood into adulthood. *J. Neurosci.* **31**, 10937–10947 (2011).

48. Vandermosten, M., Boets, B., Wouters, J. & Ghesquiere, P. A qualitative and quantitative review of diffusion tensor imaging studies in reading and dyslexia. *Neurosci. Biobehav. Rev.* **36**, 1532–1552 (2012).

49. Lebel, C. et al. Diffusion tensor imaging correlates of reading ability in dysfluent and non-impaired readers. *Brain Lang.* **125**, 215–222 (2013).

50. Schwarz, G. Estimating the Dimension of a Model. *Ann. Stat.* **6**, 461–464 (1978).

51. Neath, A. A. & Cavanaugh, J. E. The Bayesian information criterion: background, derivation, and applications. *Wiley Interdiscip. Rev. Comput. Stat* **4**, 199–203 (2012).

52. Eden, G. F. et al. Neural changes following remediation in adult developmental dyslexia. *Neuron* **44**, 411–422 (2004).

53. Shaywitz, B. A. et al. Development of left occipitotemporal systems for skilled reading in children after a phonologically- based intervention. *Biol. Psychiatry* **55**, 926–933 (2004).

54. Hedeker, D. & Gibbons, R. D. *Longitudinal Data Analysis* (John Wiley & Sons, Inc., Hoboken, New Jersey, 2006).

55. Barquero, L. A., Davis, N. & Cutting, L. E. Neuroimaging of reading intervention: a systematic review and activation likelihood estimate meta-analysis. *PLoS ONE* **9**, e83668 (2014).

56. Boltzmann, M., Mohammadi, B., Samii, A., Munte, T. F. & Russeler, J. Structural changes in functionally illiterate adults after intensive training. *Neuroscience* **344**, 229–242 (2017).

57. Kanai, R. & Rees, G. The structural basis of inter-individual differences in human behaviour and cognition. *Nat. Rev. Neurosci.* **12**, 231–242 (2011).

58. Van Beek, L., Ghesquiere, P., Lagae, L. & De Smedt, B. Left fronto-parietal white matter correlates with individual differences in children's ability to solve additions and multiplications: a tractography study. *Neuroimage* **90**, 117–127 (2014).

59. Matejko, A. A., Price, G. R., Mazzocco, M. M. & Ansari, D. Individual differences in left parietal white matter predict math scores on the Preliminary Scholastic Aptitude Test. *Neuroimage* **66**, 604–610 (2013).

60. Navas-Sanchez, F. J. et al. White matter microstructure correlates of mathematical giftedness and intelligence quotient. *Hum. Brain. Mapp.* **35**, 2619–2631 (2014).

61. Jolles, D. et al. Plasticity of left perisylvian white-matter tracts is associated with individual differences in math learning. *Brain. Struct. Funct.* **221**, 1337–1351 (2016).

62. Sykova, E. & Nicholson, C. Diffusion in brain extracellular space. *Physiol. Rev.* **88**, 1277–1340 (2008).

63. Walhovd, K. B., Johansen-Berg, H. & Karadottir, R. T. Unraveling the secrets of white matter--bridging the gap between cellular, animal and human imaging studies. *Neuroscience* **276**, 2–13 (2014).

64. Mezer, A. et al. Quantifying the local tissue volume and composition in individual brains with magnetic resonance imaging. *Nat. Med.* **19**, 1667–1672 (2013).

65. Yeatman, J. D., Wandell, B. A. & Mezer, A. A. Lifespan maturation and degeneration of human brain white matter. *Nat. Commun.* **5**, 4932 (2014).

66. Weiskopf, N., Mohammadi, S., Lutti, A. & Callaghan, M. F. Advances in MRI-based computational neuroanatomy: from morphometry to in-vivo histology. *Curr. Opin. Neurol.* **28**, 313–322 (2015).

67. Richlan, F. Developmental dyslexia: dysfunction of a left hemisphere reading network. *Front. Hum. Neurosci.* **6**, 120 (2012).

68. Travis, K. E., Ben-Shachar, M., Myall, N. J. & Feldman, H. M. Variations in the neurobiology of reading in children and adolescents born full term and preterm. *Neuroimage Clin.* **11**, 555–565 (2016).

69. Duara, R. et al. Neuroanatomic differences between dyslexic and normal readers on magnetic resonance imaging scans. *Arch. Neurol.* **48**, 410–416 (1991).

70. Rumsey, J. M. et al. Corpus callosum morphology, as measured with MRI, in dyslexic men. *Biol. Psychiatry* **39**, 769–775 (1996).

71. Bell, N. *Seeing Stars* (Gander, San Luis Obispo, CA, 2007).

72. Caruyer, E., Lenglet, C., Sapiro, G. & Deriche, R. Design of multishell sampling schemes with uniform coverage in diffusion MRI. *Magn. Reson. Med.* **69**, 1534–1540 (2013).

73. Andersson, J. L., Skare, S. & Ashburner, J. How to correct susceptibility distortions in spin-echo echo-planar images: application to diffusion tensor imaging. *Neuroimage* **20**, 870–888 (2003).

74. Andersson, J. L. & Sotiropoulos, S. N. An integrated approach to correction for off-resonance effects and subject movement in diffusion MR imaging. *Neuroimage* **125**, 1063–1078 (2016).

75. Jensen, J. H., Helpern, J. A., Ramani, A., Lu, H. & Kaczynski, K. Diffusional kurtosis imaging: the quantification of non-gaussian water diffusion by means of magnetic resonance imaging. *Magn. Reson. Med.* **53**, 1432–1440 (2005).

76. Garyfallidis, E. et al. Dipy, a library for the analysis of diffusion MRI data. *Front. Neuroinform.* **8**, 8 (2014).

77. Yendiki, A., Koldewyn, K., Kakunoori, S., Kanwisher, N. & Fischl, B. Spurious group differences due to head motion in a diffusion MRI study. *Neuroimage* **88**, 79–90 (2014).

78. Yeatman, J. D., Dougherty, R. F., Myall, N. J., Wandell, B. A. & Feldman, H. M. Tract profiles of white matter properties: automating fiber-tract quantification. *PLoS ONE* **7**, e49790 (2012).

79. Tournier, J. D., Calamante, F., Gadian, D. G. & Connelly, A. Direct estimation of the fiber orientation density function from diffusion-weighted MRI data using spherical deconvolution. *Neuroimage* **23**, 1176–1185 (2004).

80. Ben-Shachar, M., Dougherty, R. F. & Wandell, B. A. White matter pathways in reading. *Curr. Opin. Neurobiol.* **17**, 258–270 (2007).

## Acknowledgements

This work was funded by NSF BCS-1551330 to J.D.Y. We would like to thank Emily Kubota for assistance with data collection and Bruce McCandliss for feedback and comments on the manuscript.

## Author contributions

J.D.Y. and E.H. designed the study. P.M.D., J.D.Y., and E.H. collected the data. E.H., J.D.Y., and A.R. analyzed the data. E.H. and J.D.Y. wrote the paper. All authors provided feedback and edits on the manuscript.

## Additional information

**Competing interests:** The authors declare no competing interests.

