## [Peer Review File · Nature Communications]

Reviewers' comments:

Reviewer #2 (Remarks to the Author):

This manuscript examines how intensive reading instruction in children with learning disability leads to plastic changes in white matter structures. The authors provide compelling data showing that some, but not all, white matter tracts change in concert with improved reading. They also present evidence that some white matter structures have stable relationships with reading skills, while others show changing relationships over the course of the learning process.

These findings nicely extend our understanding of how white matter tracts change with reading instruction in struggling readers, and provide novel evidence that anatomy-behavior correlations can change dynamically with experience. The authors have made several improvements to the manuscript since their previous submission. They now provide a stronger motivation that is better contextualized in the broader literature on white matter plasticity. They have also added several analyses that provide greater clarity on the nature of their findings.

Below are some further suggestions and clarification the authors should address.

Major points:

1. The organization and order of the results could be modified to improve the flow. For instance, the authors nicely layout a summary of their findings at the beginning of the Discussion that presents all of the intervention-related effects followed by a summary of their findings related to changes in anatomy-behavior correlations. Best to modify the order of the Results section to mirror this organization (intervention effects first, followed by individual differences findings). As currently written, the results switch from intervention effects, to individual differences, and then back to intervention-related findings. Relatedly, it seems as though the paragraph on "normalization" of white matter (beginning "One possible interpretation of group differences..") does not fit in the section on anatomy-behavior correlations. The authors should consider placing these findings (as well as Figure 3) in a different section, or have a separate sub-section for these results.
2. Figure 3 is a very nice addition to the manuscript. However, it is not clear as to why WJ Basic reading score is used as the metric of reading, while in other analyses and figures other measures of reading ability used (e.g. combined measure of reading skills). Could the authors include another set of 3 panels using the Towre scores, to provide a more complete picture; or remain consistent with the use of the combined reading score. Also, Figure 3 uses only a sub-set of the controls (for good reason) but it is a bit confusing. More information on this would be help. Best to refer to them in the image as a subset and to indicate the sample size in the figure legend. Lastly, this figure reveals that MD has maximal change from Time 1 to Time 2, with Time 3 representing a small shift back, and recovery to Time 2 levels by Time 4. This is the case for all three measures. What is this telling us?
3. In the section "Intensive intervention changes reading skills and white matter tissue properties" the authors provide a nice analysis of the anatomical changes in the intervention and control groups. It is equally important to characterize the changes in behavior (i.e. reading skills) in order to have a more complete understanding of the overall findings. Presently, there is a general pre-post intervention comparison. Could the authors provide a behavioural analysis that mirrors the neuroimaging analysis to illustrate whether there is a group-by-session interaction in performance on the Woodcock-Johnson and TOWRE? This would allow the reader to understand whether there are changes in performance between sessions in the intervention group, but not in the control group. Relatedly, could the authors also add the standard scores from the Woodcock-Johnson and TOWRE to Figure S3, either in the same graph or in a new one. This would help visualize

intervention-related changes in performance and, if the non-intervention controls are added, then this would provide a clear overall picture.

4. To what degree do these results generalize? Typical readers may show vastly different changes in response to intensive reading instruction, because, for example, children with learning disabilities may be more malleable than typical children. Typical children may have few (or completely different) changes to white matter structures. Could the authors provide some discussion of whether they think their findings generalize to learning in typically developing children, and acknowledge the possibility that these findings are specific to children with reading impairments.

5. It seems as though the diffusion metrics have a more variability in the posterior CC than the ILF and AC. Is this the case, do the authors think that greater variability of measurement in the posterior CC could be masking any intervention-related effects? Could it be that the estimates of the posterior CC are less precise? This is an important point to address as it would alter the interpretation of the results substantially.

6. The authors present correlations between white matter and reading change over the course of instruction. The precise mechanisms for such changes remain elusive. Could the authors please provide a greater explanation for the changing correlations in the Discussion. Have the authors explored whether changes in variability to either reading scores or MD at each time point might be resulting in the changing strength of the correlations, or is variability relatively stable? Perhaps children are becoming more similar in anatomy and behavior following intervention and it is this reduction in variability that is changing the strength of the correlations.

Minor Points:

1. The authors provide the mean and standard deviation of age in both the intervention and control groups. It would also be informative to add the age-range since we know that white matter is undergoing rapid changes in childhood (Lebel et al. 2011, *Journal of Neuroscience*).

2. Could the authors please provide a citation for this statement in the introduction: "... white matter properties are often held to underlie variation in performance and causally influence individual learning trajectories."

3. It is important to acknowledge Keller and Just (2009) in the introduction. While the present manuscript expands on previous work in several ways, Keller & Just (2009) was the first papers to demonstrate that white matter properties can change in response to educational environments, specifically reading.

4. In Figure 1, could the MD panel (Fig 1B) come first since this is how the results are largely reported (MD then FA).

5. In Figure 2, could the authors add p-values to their correlations at Time 2, 3, and 4, just like in Figure S4.

6. In Figure 4 b and d, it is not entirely clear what "Change in ILF MD" and "Change in AF MD" are referring to. Is this a metric of change across all 4 time points? Could the author provide greater clarity in the caption on what is meant by these axes.

7. At the beginning of the section on "White matter properties of the AF and ILF change in concert and track with learning" it is initially unclear whether the analyses are being conducted on the intervention group, or both groups. A brief statement on which subjects are being analyzed at the beginning would improve the clarity.

8. In text, the authors refer to Figure 4a in the passage "...the time-courses of change in the AF and ILF were highly correlated for both MD and FA (MD: $r = 0.86$, $p < 0.001$, see Figure 4a; FA: $r = 0.50$, $p = 0.021$)". However, it seems as though they are actually referring to Figure 4b.

9. In the abstract the authors state: "These results underscore the importance of considering recent educational history when interpreting cross-sectional anatomy-behavior correlations." The precise implications of this sentence are unclear, and it seems to reach beyond the reported findings. Best to remove it.

10. In the introduction the authors state: "... but which could nonetheless be used for early identification of individuals in need of extra educational support 17". There has yet to be a study that demonstrates white matter properties could be used for early identification at a single-subject level. Therefore best to remove this part of the sentence to avoid confusion.

11. In the discussion the authors state "...Given the lack of a group-level intervention effect in the left arcuate in 65..." However, earlier in the paragraph the authors say "... plasticity within the left AF has recently been linked to gains in math skills following math intervention 65". These two statements seem to be in conflict, and it is not clear as to whether Jolles et al. did or did not find intervention effects in the AF. Could the authors please clarify?

Reviewer #3 (Remarks to the Author):

The authors have provided quite satisfactory responses to my concerns regarding the initial submission of this manuscript.

Reviewer #4(Remarks to the Author):

This is a longitudinal study following 24 children with reading difficulties before and during an intensive intervention. The authors measure fractional anisotropy and mean diffusivity across 4 time points, focusing on tracts with a priori associations with reading, and also exploring the full set of major tracts. They report changes in the ILF and AF that are specific to the intervention group, and relate these changes to reading ability.

I am quite skeptical about these results, due to major concerns about the analyses used. Should the authors find similar results after the following analytical requests, I would be much more convinced of these reported effects.

You did not report whether the non-intervention group improved in reading or math scores. I assume that they did not but it should be reported alongside the behavioral changes of the intervention group.

Fig1 - I would like to see many different aspects of these data: 1) in this and all subsequent cases, please add dots for each individual subject to each bar, and connect them across bars, so that the reader can see how reliable the effect is across subjects; 2) since rate of change is often more informative than difference scores, please reproduce this figure using the ratio of each session's FA and MD to the baseline score; 3&4) the reader will greatly gain from seeing session-to-session differences and rates of change – please reproduce this figure showing these metrics between sessions as opposed to comparisons with the baseline metric.

Fig1 also makes clear that the changes from baseline mostly occur right away, and remain static across sessions. Is there a commensurate change in behavior? You address this later but I am not satisfied with this approach and would rather see a simple analysis relating the subjectwise differences and rates of change from session 1 to session 2 with each reading metric. Do subjects that change the most in reading ability also have the largest changes in FA/MD between these sessions?

Similarly, this could have easily been a pre vs post study before and after intervention, but the authors admirably recorded data throughout intervention. If it had been the former, you would have simply related change (post-pre or post/pre) in reading ability with change in WM metrics. Please use FA/MD and all reading metrics, and not mean-centered data.

Each of the linear models that use session as a categorical variable is completely inappropriate. Especially given how critical session 1 is (your intercept), you cannot treat session number as fixed effect categorical variable. You are studying change in WM metrics and reading skills over time, in order to ask whether the change over time (slope) is different between control and treatment groups. Time must be used as a continuous covariate in all of these models.

Especially since there is a large difference in N between your groups (for most of the controls were only scanned once or twice, 7 were scanned 4 times compared to 21 intervention), it is important, in all cases, to test differences between groups directly. Just because you observe a change in one group and not another does not imply that they are different. Each of these models should contrast the modeled slopes over time between groups.

You need to show main effects for control group model of reading vs MD/FA. Only the interaction with session is reported. And this should be with respect to time as a continuous covariate anyways.

Fig3 - how many control subjects are included in the gray ellipse? You mention in Methods that 9 control subjects were below 85, and here you are using subjects that are within 1sd from the expected mean (i.e. were any above 115 that are also being excluded)?

In all cases, do not normalize the data, as reading scores, MD, and FA are meaningful values. If anything, you would subtract out the baseline, not the mean, as you are looking for changes with respect to a clear baseline.

Fig4 – A previous reviewer also asked about these change scores. I would like to see these same data 1) as raw (not mean-centered) data, and 2) as the difference or ratio between subsequent timepoints. Also, why are only 20 subjects shown out of 24? Please include all subjects and interpolate across missing or excluded timepoints.

You say in the paper: "we fit a linear mixed effects model to all subjects' mean-centered diffusion measurements over all time points (and use Pearson correlations as an index of effect size)." This needs a better explanation. Are you e.g. fitting ILF MD against regressors for each AF session? If so, which session of ILF MD is being fit? How are you fitting all time points against all time points? Are you collapsing sessions together? Is it a multivariate regression (ie ILF for all time points against AF for all time points)? And is the reported Pearson correlation between timepoint differences? How was this computed?

There is something weird going on with number of subjects and sessions reported

-For Fig4 why are only 20 subjects shown?

-Why are the degrees of freedom so high for the behavioral measures (77 from 24 subjects pre and post)? Also what specific test is this tscore referring to? Since it is a linear mixed model I would expect an Ftest, unless you are testing a specific contrast of model coefficients. And why are the DoF different across tests (77 for WJ basic reading and TOWRE, but 76 for WJ reading fluency, 63 for WJ calculation, and 68 for WJ math fact fluency)? If subjects were excluded for motion during MRI, please justify excluding them from behavioral measures as well.

-The DoF reported for the findings of Fig1 are (3,76) which implies 80 observations across the 4 sessions. You claim that of the 93 intervention data points from the intervention group, and you excluded 9, leaving 84 observations. Please explain, considering that this works out for the reported control data (52 sessions excluding 3 = 49, and reported DoF is (3,45) = 49 observations).

-Please explain the DoF for each test, accounting for all covariates, number of subjects and data points included.

-Why only 16 filled dots in the large scatter plots of Fig2? FigS1 makes it clear that there should be 22 subjects with data from session 1. Why are the p values in Fig2 consistent with Pearson correlations from 20 subjects?

What data were used to for the PCA for "reading skill"? Which groups were included and what time points? It should be performed on the full dataset before intervention, and the transform then applied to each subsequent session so that it is not biased towards variability in each session for

which the resulting PCA scores will be used. What are the resulting PCA coefficients so the reader can know which measure contribute most? Indeed it is a little strange to produce a generalized score based on the hyperplane that contains the highest variance across your subjects, but since it correlates so well with the composite scores it seems legitimate. Please also include all analyses that use "reading skill" but instead using each of the WJ and TOWRE metrics instead. The authors state that they perform this analysis to avoid multiple comparison issues, but since it is only 4 metrics, it truly does not impact a bonferroni correction by very much at all.

—The supplementary results included for reviewer 2, issue 3 is not a satisfactory response. There are many more analyses in the text that use "reading skill". Please perform all analyses with each reading index.

I strongly agree with previous reviewer 1, issue 4. An enormous interest in this manuscript hinges on individual subject results. Can you predict effectiveness of intervention from session 1 data? Do changes in tract metrics predict changes in reading ability for an individual subject? You have 21 subjects with 4 time-points, and so this is unquestionably feasible. I do not accept that this should be "salami-chopped" into a separate publication and is certainly more appropriate for this manuscript than the analyses in Fig5. Even a leave-one-subject-out cross validation scheme for predicting individual subject results should be included if at all possible, and would immensely strengthen this paper.

The authors state in Methods that mean, radial, and axial diffusivity (MD, RD, AD) and fractional anisotropy (FA) were mapped onto each tract to create a "Tract Profile". Why are these metrics not reported when a) they must have been computed in order to produce FA and MD, and b) they have been previously reported to relate to reading ability (e.g. Yeatman et al., 2011) and dyslexia (e.g. Vandermosten et al., 2012)?

Minor corrections:

FigS1a should say RMS displacement < (not >) 0.7mm

FigS2 - *between (sp.)

FigS4 - *reading (sp.)

Reviewers' comments:

Reviewer #2 (Remarks to the Author):

This manuscript examines how intensive reading instruction in children with learning disability leads to plastic changes in white matter structures. The authors provide compelling data showing that some, but not all, white matter tracts change in concert with improved reading. They also present evidence that some white matter structures have stable relationships with reading skills, while others show changing relationships over the course of the learning process.

These findings nicely extend our understanding of how white matter tracts change with reading instruction in struggling readers, and provide novel evidence that anatomy-behavior correlations can change dynamically with experience. The authors have made several improvements to the manuscript since their previous submission. They now provide a stronger motivation that is better contextualized in the broader literature on white matter plasticity. They have also added several analyses that provide greater clarity on the nature of their findings.

Thank you.

Below are some further suggestions and clarification the authors should address.

Major points:

1. The organization and order of the results could be modified to improve the flow. For instance, the authors nicely layout a summary of their findings at the beginning of the Discussion that presents all of the intervention-related effects followed by a summary of their findings related to changes in anatomy-behavior correlations. Best to modify the order of the

Results section to mirror this organization (intervention effects first, followed by individual differences findings). As currently written, the results switch from intervention effects, to individual differences, and then back to intervention-related findings. Relatedly, it seems as though the paragraph on “normalization” of white matter (beginning “One possible interpretation of group differences..”) does not fit in the section on anatomy-behavior correlations. The authors should consider placing these findings (as well as Figure 3) in a different section, or have a separate sub-section for these results.

Our previous organization kept all of the individual difference findings (pre-intervention, and the effect of intervention on anatomy-behavior correlations) together, which required some switching between pre-intervention individual differences and intervention effects. We have taken the reviewer’s suggestion, and we now report pre-intervention individual differences first, in a section titled “Tracts connecting the core reading circuitry correlate with pre-intervention reading skill”. We follow this with a summary of intervention-driven effects, including the influence of intervention on observed anatomy-behavior relationships. We also present the paragraph on “normalization” of white matter differences in a separate section (“Reading intervention does not ‘normalize’ differences in the white matter”). We agree that these changes improve the flow and thank the reviewer for this suggestion.

2. Figure 3 is a very nice addition to the manuscript. However, it is not clear as to why WJ Basic reading score is used as the metric of reading, while in other analyses and figures other measures of reading ability used (e.g. combined measure of reading skills).

For Figure 3, we chose to plot WJ Basic Reading, which is a standardized measure, to let the reader visualize cross-sectional and intervention-driven effects with reference to an age-normed, population mean. Thus, while we prefer to summarize the various reading measures using the Reading Skill composite elsewhere in the manuscript, we find the use of a specific normed measure to be the most intuitive way to show intervention-drive growth relative to typical readers.

To clarify this decision, we have revised the text in the Results section (p. 9) as follows:

Figure 3 shows changes in MD and reading scores for the intervention group, relative to the subset of non-intervention controls who had reading skills in the typical range. We defined ‘Typical Readers’ as Control Group subjects with timed (TOWRE Index) and untimed (WJ Basic Reading Score) reading accuracy within a standard deviation of the population mean (at or above 85 on both measures). For the intervention group, we plot changes in both WJ Basic Reading and the TOWRE Index (rather than composite Reading Skill), in order to situate the cross-sectional and intervention-driven effects relative to an age-normed, population mean.

Could the authors include another set of 3 panels using the Towre scores, to provide a more complete picture; or remain consistent with the use of the combined reading score.

We have added a set of 3 panels using the TOWRE scores. As shown below, these data mirror the effects shown for WJ BRS. We agree that this provides a more complete picture, and thank the reviewer for this suggestion.

Figure 3. Reading intervention does not ‘normalize’ differences in the white matter. Reading skills are plotted as function of mean diffusivity for each session (1-4) for the left arcuate, ILF, and posterior callosal tracts in the intervention group. The gray ellipse in each panel shows the mean and standard error for the subset non-intervention control subjects with reading skills in the typical range (poor reading controls were excluded, leaving $n = 10$ typical reading controls). The dashed gray arrow shows the expected trajectory for MD values if the intervention group were becoming more similar to the typical reading controls in terms of both reading skills and MD. In contrast, the true trajectory of change in plotted as a colored arrow in each panel. The intervention group includes some readers with only moderate reading impairments (and, therefore, higher MD values), and so the group difference in pre-intervention scores is less than would be expected for a group of good vs. poor readers.

Also, Figure 3 uses only a sub-set of the controls (for good reason) but it is a bit confusing. More information on this would be help. Best to refer to them in the image as a subset and to indicate the sample size in the figure legend.

Figure 3 includes the 10 control subjects whose TOWRE Index and WJ BRS scores fell within one standard deviation of the population mean (greater than or equal to 85) at Session 1. We have clarified the sample size in the figure legend (see above), and we have labeled that portion of the image “Typical Readers”. We hope that this differentiates these subjects from the larger “Non-Intervention” control group.

Lastly, this figure reveals that MD has maximal change from Time 1 to Time 2 , with Time 3 representing a small shift back, and recovery to Time 2 levels by Time 4. This is the case for all three measures. What is this telling us?

As shown in **Figure 5**, the time-course of change varies across individual subjects. Some subjects show most change immediately, while others show greater change during the last two measurement sessions. At the group level, we see a significant effect between sessions 1 and 2, relative stability between sessions 2 and 3, and some additional change between sessions 3 and 4. The shift back between session 2 and 3 is not statistically significant, so we did not want over interpret this effect. But we do agree with the reviewer that future work, with a larger sample size, should follow up on potential non-linearities and individual differences in the time-course of plasticity.

3. In the section “Intensive intervention changes reading skills and white matter tissue properties” the authors provide a nice analysis of the anatomical changes in the intervention and control groups. It is equally important to characterize the changes in behavior (i.e. reading skills) in order to have a more complete understanding of the overall findings. Presently, there is a general pre-post intervention comparison. **Could the authors provide a behavioural analysis that mirrors the neuroimaging analysis to illustrate whether there is a group-by-session interaction in performance on the Woodcock-Johnson and TOWRE?** This would allow the reader to understand whether there are changes in performance between sessions in the intervention group, but not in the control group.

We thank the reviewer for this suggestion. We have added this analysis to the Results section, p. 4. In addition to the WJ and TOWRE reading tests, we test this interaction for the math skills measures. One intricacy that is revealed by this analysis is that, among the strong reading control subjects, we observe a practice effect for timed reading tests. Meanwhile, among the dyslexic control subjects, all test scores are stable, showing no change. Thus for a reading matched control group, the group by time interactions are significant for all reading scores (but not for the math scores). When the full sample of controls is used (including both good and poor readers), we see an interaction for the WJ Basic Reading composite (an untimed measure of reading accuracy). All these findings are now referenced in the text and the relevant statistics are reported either in the Results section (p. 5) and Supplementary Table 2. Additionally, main

effects for the intervention group, the non-intervention controls, and the reading-matched controls are plotted in Supplementary Figure 2B.

Relatedly, could the authors also add the standard scores from the Woodcock-Johnson and TOWRE to Figure S3, either in the same graph or in a new one. This would help visualize intervention-related changes in performance and, if the non-intervention controls are added, then this would provide a clear overall picture.

We have added a set of panels to Figure S1 (previously Figure S3) showing intervention effects in each of the WJ and TOWRE subtests. To provide the most complete pictures, we include both the reading and math measures. For consistency with the rest of the manuscript, we plot coefficients from the linear-mixed effects model used to quantify intervention effects throughout. As shown in Figure S1B, intervention subjects improve in all of the reading, but not the math, measures. Reading matched control subjects do not improve in any measure. Within the full group of controls, which includes the skilled reading subset, we observe moderate improvements in the Reading Fluency measure (which we attribute to practice effects), while the remaining reading measures show no statistical difference over the course of 8 weeks. Again, we agree that this presentation of the results strengthens the manuscript, and further highlights the specificity of learning effects to the intervention group.

A.

B.

Figure S1. Intervention driven growth in reading performance. (A) We created a single summary index of reading skills based on conducting principal component analysis of the Woodcock Johnson and Test of Word Reading Efficiency standard scores (see Methods). Intervention driven change in this Reading Skill composite is plotted as a function of intervention hours and shows highly significant change (linear mixed effects model, fixed effect of intervention hours and random effect of subject, $p < 10^{-9}$). (B-C) Linear growth in each of the standardized reading measures comprising the Reading Skill composite. In the intervention group, each of the reading subtests grew significantly during the intervention.

4. To what degree do these results generalize? Typical readers may show vastly different changes in response to intensive reading instruction, because, for example, children with learning disabilities may be more malleable than typical children. Typical children may have few (or completely different) changes to white matter structures. Could the authors provide some discussion of whether they think their findings generalize to learning in typically developing children, and acknowledge the possibility that these findings are specific to children with reading impairments.

We agree that this is an important point, although not one that we can address directly with the current dataset. In the Discussion section, we note that intervention-driven effects extend to anatomical regions not typically associated with the core reading circuitry. Indeed, we suggest that similar effects might emerge during a sufficiently intensive intervention targeting math, or other cognitive skills. Although it would not be feasible to enroll skilled readers in a highly intensive reading intervention program, future work examining the generalizability of these effects in other domains, such as math, would allow an examination of general learning effects in a broader population and should help clarify the role of domain specific deficits. We now discuss this point made by the reviewer in the relevant section of the Discussion.

5. It seems as though the diffusion metrics have a more variability in the posterior CC than the ILF and AC. Is this the case, do the authors think that greater variability of measurement in the posterior CC could be masking any intervention-related effects? Could it be that the estimates of the posterior CC are less precise? This is an important point to address as it would alter the interpretation of the results substantially.

Estimates for the posterior CC are highly reliable across measurement sessions. For the intervention group, the mean intraclass correlation across all pairs of sessions (i.e., Session 1 vs. 2, 2 vs. 4, etc.) is 0.93 (standard deviation = 0.012). This is excellent reliability and rivals the test-retest reliability of most standardized behavioral measures. Thus, we are confident that CC diffusion metrics are estimated reliably within a subject. This would permit detection of session-by-session changes within a subject, should they exist. We do acknowledge that MD and FA

estimates in the posterior CC do show greater between-subject variability than the other two tracts; however, we account for this between-subject variance statistically through use of a linear mixed effects model.

6. The authors present correlations between white matter and reading change over the course of instruction. The precise mechanisms for such changes remain elusive. Could the authors please provide a greater explanation for the changing correlations in the Discussion. **Have the authors explored whether changes in variability to either reading scores or MD at each time point might be resulting in the changing strength of the correlations, or is variability relatively stable?** Perhaps children are becoming more similar in anatomy and behavior following intervention and it is this reduction in variability that is changing the strength of the correlations.

This is an important point. In fact, because we see learning effects across the range of initial skill levels (i.e., the distribution of reading scores shifts, rather than being highly compressed, as would be the case if only the subset of the most impaired readers showed learning effects), the variance across time points is relatively stable. Below, we quantify the stability of variances across time points for each of the core white matter tracts shown in Figure 2, and the for Reading Skill composite. We now include this analysis as **Supplementary Table 4**.

	Reading Composite	Posterior CC	Left ILF	Left Arcuate
Session 1 vs. 2	$F(19,19)=1.088,$ $p > 0.05$	$F(19,19)=1.17,$ $p > 0.05$	$F(19,19)=1.045,$ $p > 0.05$	$F(19,19)=1.16,$ $p > 0.05$
Session 1 vs. 3	$F(19,19)=1.24,$ $p > 0.05$	$F(19,19)=1.30,$ $p > 0.05$	$F(19,19)=1.25,$ $p > 0.05$	$F(19,19)=1.075,$ $p > 0.05$
Session 1 vs. 4	$F(19,19)=1.49,$ $p > 0.05$	$F(19,19)=1.70,$ $p > 0.05$	$F(19,19)=1.98,$ $p > 0.05$	$F(19,19)=2.27,$ $p > 0.01$

Table 1. F-statistics in each cell represent the ratio of variance across time points for each white matter tract, and the Reading Skill composite, calculated with the larger of the 2 variances in the numerator. Statistics were computed using the 20 intervention subjects with the full set of 4 MRI data points.

For the left arcuate, the variance is greater at Session 4 than Session 1, although this is a moderately sized effect, which holds at $p < 0.05$, but not $p < 0.01$. In any case, if changing correlations were driven solely by changes in relative variance, the increased variance should not produce an *attenuated* correlation at time point 4. Moreover, the variance across sessions 1-3 is not statistically different, and so the dramatic changes in the correlation coefficient for this pathway (which occurs between sessions 1 and 2) cannot be attributed simply to differences in relative variance over time. Similarly, variance in the posterior CC, left ILF, and Reading

Composite are well matched across sessions. We have added the following text to the Results section (p. 11):

*As shown in **Supplementary Table 4**, there was no statistical difference in variance across sessions (indeed, variances were nearly matched; see also **Figure 3**, which plots means and errors for each session). Therefore, changing anatomy-behavior correlations were not driven by differences in relative variance over time, and instead reflect learning related dynamics in the reading and white matter measures.*

Minor Points:

1. The authors provide the mean and standard deviation of age in both the intervention and control groups. It would also be informative to add the age-range since we know that white matter is undergoing rapid changes in childhood (Lebel et al. 2011, Journal of Neuroscience).

Thank you for this suggestion. We have added the age-range (7-12 years) to the Methods section (*Participants*, p. 18).

2. Could the authors please provide a citation for this statement in the introduction: "... white matter properties are often held to underlie variation in performance and causally influence individual learning trajectories.

We thank the reviewer for noting this omission. We have added relevant citations to this statement (Introduction, p. 2).

3. It is important to acknowledge Keller and Just (2009) in the introduction. While the present manuscript expands on previous work in several ways, Keller & Just (2009) was the first papers to demonstrate that white matter properties can change in response to educational environments, specifically reading. (added p.2)

We agree that this is a key citation. We previously cited Keller and Just (2009) in the Discussion section, but we now also cite this work in the Introduction (p.2). We agree that this helps contextualize the current study.

4. In Figure 1, could the MD panel (Fig 1B) come first since this is how the results are largely reported (MD then FA).

We appreciate this suggestion and have revised the figure accordingly, as shown below:

Figure 2. Change versus stability in Tract Profiles during reading intervention. (A) Mean diffusivity values were mapped onto each of 100 evenly spaced nodes spanning termination points at the gray-white matter boundary to create a ‘Tract Profile’ (see Methods and⁶¹ for additional details of this analysis). For visualization purposes, the middle 80 nodes are plotted. Each line represents the group average mean diffusivity (MD) across subjects, measured at four time-points: pre-intervention (Session 1), after ~2.5 weeks of intervention (Session 2), after ~5 weeks of intervention (Session 3), and after 8 weeks of intervention (Session 4). Shaded error bars give ± 1 standard error of the mean. Color values indicate session, ranging from darkest (Session 1) to brightest (Session 4) for each tract. The x-axis shows the location where each tract was clipped prior to analysis (corresponding to black boundary lines in renderings, above). Tract renderings are shown for an example subject. The middle 60% (bounded by black lines) of each tract was analyzed in (B-C), to avoid partial volume effects that occur at endpoints of the tract, where it enters cortex. Both the AF and inferior ILF, but not the posterior callosal connections, show a systematic decrease in MD over the course of intervention. (B-C) Bars show model predicted change (coefficients and standard errors from mixed effects model) in MD (B) and FA (C) for each session. Bar heights represent the magnitude of change observed in that session, relative to Session 1 (pre-intervention) baseline. As described in the main text, both the arcuate fasciculus (AF) and inferior longitudinal fasciculus (ILF) showed significant change between sessions for the intervention group (filled bars), but not the control group (unfilled bars). Asterisks indicate a significant decrease in MD (B) or increase in FA (C) for each session relative to the pre-intervention baseline at a Bonferroni corrected $p < 0.05$ (*) and $p < 0.01$ (**).

5. In Figure 2, could the authors add p-values to their correlations at Time 2, 3, and 4, just like in Figure S4.

We have added these p-values.

6. In Figure 4 b and d, it is not entirely clear what “Change in ILF MD” and “Change in AF MD” are referring to. Is this a metric of change across all 4 time points? Could the author provide greater clarity in the caption on what is meant by these axes.

“Change in ...” is meant to convey that the points being plotted are difference scores, relative to each subject’s mean. We agree that this terminology is imprecise, and we have revised the axes to read, e.g., “ILF MD (mean centered)”. This also matches the main text of the manuscript and is explained in the figure caption.

7. At the beginning of the section on “White matter properties of the AF and ILF change in concert and track with learning” it is initially unclear whether the analyses are being conducted on the intervention group, or both groups. A brief statement on which subjects are being analyzed at the beginning would improve the clarity.

We have revised the text so that we refer to “intervention subjects” rather than “subjects” for analysis carried out specifically within the intervention group. We also introduce this section by asking “whether changes in the AF and ILF occur in synchrony *in the intervention group*” (Results, p.9, emphasis added).

8. In text, the authors refer to Figure 4a in the passage “...the time-courses of change in the AF and ILF were highly correlated for both MD and FA (MD: $r = 0.86$, $p < 0.001$, see Figure 4a; FA: $r = 0.50$, $p = 0.021$)”. However, it seems as though they are actually referring to Figure 4b.

The time courses are plotted, in different forms, in both 4a and 4b. We now simply refer to figure 4, to avoid confusion.

9. In the abstract the authors state: “These results underscore the importance of considering recent educational history when interpreting cross-sectional anatomy-behavior correlations.” The precise implications of this sentence are unclear, and it seems to reach beyond the reported findings. Best to remove it.

We have revised this sentence to read, “These results underscore the importance of considering recent experience when interpreting cross-sectional anatomy-behavior correlations.” We hope that this captures the effects reported in the manuscript (i.e., cross-

sectional correlations fundamentally change over the time-scale of weeks), without reaching more broadly toward “educational history” as a moderating factor.

10. In the introduction the authors state:“... but which could nonetheless be used for early identification of individuals in need of extra educational support 17”. There has yet to be a study that demonstrates white matter properties could be used for early identification at a single-subject level. Therefore best to remove this part of the sentence to avoid confusion.

While we wish to acknowledge the possibility that stable anatomical differences could have diagnostic value, since the aim of identifying biomarkers underlies much of the current work in the field, we agree that it is important to avoid implying that white matter properties can already provide accurate single-subject diagnosis. We have therefore revised the sentence to read “...but which could *plausibly* be used for early identification of individuals in need of extra educational support” (emphasis added). We hope that this avoids creating confusion.

11. In the discussion the authors state “...Given the lack of a group-level intervention effect in the left arcuate in 65...” However, earlier in the paragraph the authors say “... plasticity within the left AF has recently been linked to gains in math skills following math intervention 65”. These two statements seem to be in conflict, and it is not clear as to whether Jolles et al. did or did not find intervention effects in the AF. Could the authors please clarify?

In the manuscript cited, the authors carried out an analysis of individual differences, although they did not detect a group level intervention effect. We have revised the Discussion to clarify their findings, and their possible relationship to our study (p. 16):

Interestingly, individual differences in plasticity within the left AF have recently been linked to individual gains in math skills following math intervention⁶⁸, even though the left AF is conventionally associated with language related skills. It should be noted, however, that in⁶⁸, math skills training was not associated with a significant change in the arcuate at a group level, and therefore the previous set of findings differ from ours. [...] Despite the lack of a group-level intervention effect in the left arcuate in⁶⁸, it remains possible that a sufficiently intense math intervention might prompt changes not only within the left arcuate, but within many of the same tracts identified here.

Reviewer #3 (Remarks to the Author):

The authors have provided quite satisfactory responses to my concerns regarding the initial submission of this manuscript.

Thank you.

Reviewer #4(Remarks to the Author):

This is a longitudinal study following 24 children with reading difficulties before and during an intensive intervention. The authors measure fractional anisotropy and mean diffusivity across 4 time points, focusing on tracts with a priori associations with reading, and also exploring the full set of major tracts. They report changes in the ILF and AF that are specific to the intervention group, and relate these changes to reading ability.

I am quite skeptical about these results, due to major concerns about the analyses used. Should the authors find similar results after the following analytical requests, I would be much more convinced of these reported effects.

We hope that by clarifying our analytical choices we can alleviate this reviewer's concerns. We also present the requested re-analysis here, and we include these new analyses in the main manuscript or supplements where appropriate. In cases where we believe the suggested analyses to be improper, we provide our reasoning. We also release the data, and code to reproduce each figure. This will allow readers to examine our results from many different angles.

You did not report whether the non-intervention group improved in reading or math scores. I assume that they did not but it should be reported alongside the behavioral changes of the intervention group.

Reviewer 2 also requested a more detailed analysis of the behavioral measures. We agree with both reviewers that this information is important for evaluating the efficacy of the intervention. The relevant statistic is a group (intervention/control) by time (days of intervention/control period) interaction, demonstrating that the rate of change in reading skills is significantly greater for those who were enrolled in the intervention compared to those who were not. We now report this analysis at the beginning of the results section for a reading matched control group ($n = 9$), and in **Supplementary Table 2** for the full group of non-intervention control subjects. Critically, for the reading matched group, we observed a significant interaction for the reading, but not math, measures. Within the full group of controls, which includes the skilled reading subset, we see an interaction for the WJ Basic Reading composite (an untimed measure of reading accuracy). We attribute this difference to a slight increase in performance on the timed measures in the skilled reading controls (See our response to Reviewer #2 for a more complete description). We refer to all of these results in the main text (Results, p. 5):

Growth in reading skill was specific to the intervention group, as indicated by a significant group (intervention versus control) by time (days) interaction for all reading, but not math, measures using a reading-matched group of control subjects ($n = 9$). [...] For WJ Basic Reading Skills, we saw no significant effect of group ($F(1,94) = 0.16, p = 0.68$) or time ($F(1,94) = 0.19, p = 0.67$), but a significant group-by-time interaction ($F(1,94) = 4.22, p = 0.042$), indicating that growth in reading skills during the intervention period was specific to the intervention subjects.

Similarly, for the TOWRE Index, we saw no significant effect of group ($F(1,94) = 1.12, p = 0.29$) or time ($F(1,94) = 0.24, p = 0.63$), but a significant group-by-time interaction ($F(1,94) = 4.069, p = 0.047$). For the WJ Calculation test, we saw a significant main effect of group ($F(1,94) = 4.10, p = 0.046$) but not time ($F(1,94) = 0.31, p = 0.58$), and no significant group-by-time interaction ($F(1,94) = 1.13, p = 0.29$), consistent with stability of this measure in both groups. Results for the full control sample ($n = 19$) are given in **Supplementary Table 2**; this analysis shows that amongst the skilled reading control subjects, performance improved with repeated testing for the timed measures (TOWRE and Reading Fluency). In all control subjects, untimed measures (WJ Basic Reading) were stable, showing no change over 8 weeks. In other words, skilled readers benefitted slightly from repeated practice with the timed reading tests, while poor readers did not show any improvements with practice, and only showed an improvement in performance as a result of the intervention program.

Additionally, we now plot main effects for each reading and math skills subtest for the control group (both the reading matched subset, and the full sample) in **Supplementary Figure 1B**. We agree with the reviewer that this presentation of the results adds additional information to the manuscript, and increases confidence in the large reading-specific effects seen in the intervention group.

Fig1 - I would like to see many different aspects of these data: 1) in this and all subsequent cases, please add dots for each individual subject to each bar, and connect them across bars, so that the reader can see how reliable the effect is across subjects;

We already provide visualizations of individual subject data in Figure 4, which we feel is the appropriate place for these individual curves. We also feel that the current way individual subjects trajectories are displayed is a much clearer way to visualize and appreciate individual time-courses rather than superimposing all the curves on top of each other.

2) since rate of change is often more informative than difference scores, please reproduce this figure using the ratio of each session's FA and MD to the baseline score;

Figure 2 (previously, Figure 1) displays the coefficients and error bars for a linear mixed effects model, as stated in the caption. Linear mixed effects models are the most widely used approach for analyzing longitudinal data (for example see Hedeker and Gibbons, *Longitudinal Data Analysis*, Ch. 4, p. 47-48, and Verbeke and Molenberghs, *Linear Mixed Models for Longitudinal Data*, Ch. 3, p. 19-29). The reviewer asks us instead to calculate the ratio between each session's diffusion measures and the baseline measure. While we can appreciate the appeal of the proposed analysis, highly cited papers in clinical trials research argue against this approach. For example see: Vinkers and Altman, 2001, Analyzing controlled trials with baseline and follow up measurements (1205 citations); Vickers, 2001, The use of percentage change from baseline as an outcome in a controlled trial is statistically inefficient: a simulation study (261 citations). But more importantly, while the raw and mean-centered diffusion measures closely approximate a Gaussian distribution (left panel below), the ratio transform proposed by the reviewer makes

the data **highly non-Gaussian**. We cannot think of any justification for transforming normally distributed data in a manner that makes the distribution far from normal.

Histograms show raw FA values for the left arcuate and ILF (left column) and ratio transformed FA values (right column). A Shapiro-Wilk test (see testNormality.m in LMB-Diffusion) confirms that the ratio transformed values are highly non-Gaussian. The p-values associated with the Shapiro-Wilk test are displayed above each histogram.

While we disagree with the assertion that we should transform our data to ratios, we do appreciate the fact that every reader will have a different preference for data analysis and visualization. **Thus, we have decided to publicly release a database of the diffusion measures and reading scores.** The data can be downloaded from a public repository (<http://INSERT URL UPON PUBLICATION>). This will allow every reader to apply various transforms to the data and examine how this affects the results.

3&4) the reader will greatly gain from seeing session-to-session differences and rates of change – please reproduce this figure showing these metrics between sessions as opposed to comparisons with the baseline metric.

The below figure shows session-to-session differences in MD and FA for the left arcuate and ILF. This analysis falls in line with the results reported in the manuscript: A majority of change occurs between session 1 and session 2. The average change between session 2 and session 3 is near zero, although additional change is detectable between session 3 and session 4 for MD but not for FA. (This is consistent with the non-linearity of FA effects, which we now examine in greater detail in the Results section, p. 5-6). Given that this analysis is in line with the original finding, we prefer to keep our original analysis using the linear mixed effects model, since it is a widely used approach and it provides a more elegant way to deal with missing data points than simply excluding subjects (as in the analysis, below).

Session-to-Session Difference Scores. Difference scores are plotted for the left arcuate and ILF for both MD and FA. This analysis reproduces the key findings shown in Figure 2 (previously, Figure 1) in the manuscript.

Fig1 also makes clear that the changes from baseline mostly occur right away, and remain static across sessions. Is there a commensurate change in behavior?

The reviewer is correct that the changes in diffusion properties occur very rapidly. Coincident with these effects, we observe a significant change in behavior between session 1 and session 2. Additionally, there are significant changes in behavior in the later sessions - i.e., intervention subjects continue to improve in their reading scores throughout the intervention period. This finding is shown in supplemental figure **S1A**. Thus, although there is a statistically significant relationship between the time-course of white matter and behavioral changes, the relationship is

not perfectly linear over time, and there is additional unexplained variance as well. We now explicitly note this point in the Discussion.

You address this later but I am not satisfied with this approach and would rather see a simple analysis relating the subjectwise differences and rates of change from session 1 to session 2 with each reading metric.

The suggested analysis tests whether the magnitude of change in diffusion properties correlates with the magnitude of change in reading scores. This proposed analysis tests a different effect than the longitudinal analysis that we reported in the original manuscript, and we agree that it is also interesting (see below figure to highlight the difference between these analysis approaches). We have added this analysis for each reading metric in **Supplementary Table 7**. Indeed, some of the measures (e.g., the TOWRE subtests) are significantly correlated with session 1 to session 2 white matter changes. However, we prefer not to over interpret this result, since it depends on a post-hoc analysis involving multiple reading measures, some of which show more moderate relationships with session 1 to session 2 white matter changes. We therefore prefer to focus the manuscript on the original analysis, which related each subject's individual time-course of change in MD and FA to their time-course of change in reading skill. We do, however, include the session-to-session analysis in **Supplementary Table 7**, so that this information is available for the reader.

To clarify our point, the below schematic illustrates the longitudinal effect we model in the manuscript, and demonstrates how time-courses of change in behavior and diffusion can be tightly coupled, while there is no correlation between differences scores. This can reflect a lack of between-subject variance in Session 2 vs. Session 1 difference scores (A) or a lack of between-subject variance in Session 4 vs. Session 1 difference scores (A and B). There are also other ways that time-courses can be coupled without a correlation in difference scores. We are not arguing that correlations between differences scores are not interesting, we are just noting that detecting the correlation suggested by the reviewer was not our goal in collecting dense longitudinal measurements. Our goal (as stated in the manuscript) was to model whether neural and behavioral changes occur in synchrony. Detecting correlations between the overall magnitude of change would have benefited from a different design.

Modeling the correspondence between trajectories of white matter and behavioral change. Growth trajectories can be tightly coupled within individuals despite a lack of correlation between session-to-session differences scores. For example, a lack of between-subject variance in Session 2 vs. Session 1 (A) or a lack of between-subject variance in Session 4 vs. Session 1 (A and B) leads to a lack of correlation, despite close coupling between individual time courses when all time points are considered. Because we have dense longitudinal measurements, we have better statistical power to detect the coupling between time-courses than individual differences in change scores.

Do subjects that change the most in reading ability also have the largest changes in FA/MD between these sessions? Similarly, this could have easily been a pre vs post study before and after intervention, but the authors admirably recorded data throughout intervention. If it had been the former, you would have simply related change (post-pre or post/pre) in reading ability with change in WM metrics. Please use FA/MD and all reading metrics, and not mean-centered data.

Like the analysis above, the requested analysis would test whether the total magnitude of change in reading scores depends linearly on the total magnitude of change in diffusion measures. This addresses a different point than the analyses that we reported in the paper, which focused on the time-course, rather than the absolute magnitude of change. Again, we are hesitant to over interpret the relationships between the magnitude of change in these measures. However, we have added this information to **Supplementary Table 9**. In general we do not find a clear linear relationship between overall changes in MD and reading abilities over time. The absence of this relationship partially explains our observation that anatomy-behavior correlations change over time: If both measures changed at the same rate, then the cross-sectional relationship between white matter and behavior should remain intact throughout the intervention. Instead, we find a more complex relationship between the total magnitude of change in these measures, which nonetheless appear to change in synchrony within individual subjects.

Each of the linear models that use session as a categorical variable is completely inappropriate. Especially given how critical session 1 is (your intercept), you cannot treat session number as fixed effect categorical variable. You are studying change in WM metrics and reading skills over time, in order to ask whether the change over time (slope) is different between control and treatment groups. Time must be used as a continuous covariate in all of these models.

The categorical model, which includes a fixed effect of session and a random effect (intercept) of subject, is a commonly used approach which is appropriate for quantifying change across sessions, given that the testing sessions occurred at regular time intervals (Hedeker and Gibbons, *Longitudinal Data Analysis*, and Verbeke and Molenberghs, *Linear Mixed Models for Longitudinal Data*). In our case, the continuous and categorical predictors were highly correlated ($r = 0.97$ for 'session' vs. 'days', the continuous variable indexing time spent in the intervention/control period). Indeed, given that we scheduled testing sessions at regular intervals, this approach essentially amounts to a repeated measures ANOVA. A benefit of modeling session-to-session change is that it allows the reader to appreciate potential non-linearities in change over time. As can be appreciated from Figure 1, most of the change occurs between session 1 and session 2. The original paper also included an analysis in which time (hours of intervention) was treated as a continuous variable. In this revision we have also updated the statistics to include a time-by-group interaction, which substitutes "days since start of intervention (or control period)" to create a meaningful continuous predictor for each group. These statistics are now reported in Results, p.6-7. Finally, we have reorganized and revised our Results section to clarify our reasoning for using the categorical model, and to highlight the continuous time-by-group interaction. Both analyses are interesting (and hence included in the paper).

Especially since there is a large difference in N between your groups (for most of the controls were only scanned once or twice, 7 were scanned 4 times compared to 21 intervention), it is important, in all cases, to test differences between groups directly. Just because you observe a change in one group and not another does not imply that they are different. Each of these models should contrast the modeled slopes over time between groups.

We agree with the reviewer that this is a critical test, and it is one that we already provide in the manuscript (along with additional analysis of between-group differences, which we supplied in the previous response to reviews, and in supplemental material). In this revision, we have also added the analysis contrasting slopes fit to the continuous predictor (days), as described above.

You need to show main effects for control group model of reading vs MD/FA. Only the interaction with session is reported. And this should be with respect to time as a continuous covariate anyways.

We now report main effects for the analysis of time as a continuous covariate.

Fig3 - how many control subjects are included in the gray ellipse? You mention in Methods that 9 control subjects were below 85, and here you are using subjects that are within 1sd from the expected mean (i.e. were any above 115 that are also being excluded)?

Figure 3 (which was added in the previous revision) includes 10 control subjects whose Session 1 TOWRE and WJ-BRS fell within 1 standard deviation of the mean. This captured all of the “typical reading” control subjects in this sample (i.e., we are not excluding any of the skilled readers from this figure). We have added the following text to the section titled “Reading intervention does not ‘normalize’ differences in the white matter” to clarify this (Results, p. 9):

The group of ‘Typical Readers’ included all of the skilled readers in our sample (i.e., we did not exclude any skilled reading controls subjects with scores > 115).

In all cases, do not normalize the data, as reading scores, MD, and FA are meaningful values. If anything, you would subtract out the baseline, not the mean, as you are looking for changes with respect to a clear baseline.

The only case in which we report normalized data is in the analysis of individual trajectories (Figure 5, previously Figure 4). We do so to facilitate visualization of individual learning trajectories, after removing between subject differences. **Importantly, the key results hold when baseline normalized data are substituted in the analysis, as now reported in Supplementary Table 5.** However, since baseline normalization introduces heteroskedasticity, we prefer to report effects using mean centered data in the main text. Removing each subjects mean is the proper way to test whether the time-course of diffusion changes matches the time-course of reading skill changes. For a description of this statistical approach see: Hedeker and Gibbons, (2006), *Longitudinal Data Analysis*, Ch. 4, section 4.5.2.1, “Within and Between-Subjects Effects for Time-Varying Covariates”.

Fig4 – A previous reviewer also asked about these change scores. I would like to see these same data 1) as raw (not mean-centered) data, and 2) as the difference or ratio between subsequent timepoints. Also, why are only 20 subjects shown out of 24? Please include all subjects and interpolate across missing or excluded timepoints.

As noted above, we have added an analysis of difference scores and report the results in **Supplementary Table 7.**

In Figure 5 (previously Figure 4), we include the 20 subjects with 3 or more usable data sets, since we cannot accurately estimate a growth trajectory in subjects with a single, or only 2, data points. We prefer not to interpolate missing data points for visualization because we think it would be misleading, and would suggest that we had more usable data than we had. In the present study, we used strict standards to avoid data quality issues (e.g., related to subject motion) that typically arise in pediatric samples. We feel that it is important to honestly represent the amount of data that was excluded. Throughout the manuscript we note the various tests we

performed to rule out subject motion as a potential source of the reported effects. We believe that our careful handling of data quality issues is a strength of our study.

You say in the paper: "we fit a linear mixed effects model to all subjects' mean-centered diffusion measurements over all time points (and use Pearson correlations as an index of effect size)." This needs a better explanation. Are you e.g. fitting ILF MD against regressors for each AF session? If so, which session of ILF MD is being fit? How are you fitting all time points against all time points? Are you collapsing sessions together? Is it a multivariate regression (ie ILF for all time points against AF for all time points)? And is the reported Pearson correlation between timepoint differences? How was this computed?

As we note in the Results section, p. 13, data were "mean-centered to remove inter-subject differences in baseline reading ability, and a linear mixed effects model was fit to shifted (lag = -1 and lag = 1) and un-shifted (lag = 0) versions of the time-courses." In other words, we used a mixed model to predict MD (or FA) in the left ILF from MD (of FA) in the left arcuate, for all subjects. We carried out the same analysis using time-lagged versions of each tracts time course (shifted within subject, i.e., data from Subject 1, Session 1 predicting data from Subject 1, Session 2, etc.). We used the same approach to compare Reading Skill data to MD (or FA). See: Hedeker and Gibbons, (2006), *Longitudinal Data Analysis*, Ch. 4, section 4.5.2.1, "Within and Between-Subjects Effects for Time-Varying Covariates", for a description of mean-centering longitudinal data.

There is something weird going on with number of subjects and sessions reported

Unfortunately, usable MR sessions do not perfectly align with usable behavioral sessions. For example, a subject might complete all reading measures, but his/or MRI data might be unusable. A different subject, with usable MRI data, might not have had time to complete one of the reading tests. Therefore, the exact degrees of freedom associated with a given test (and the exact number of plotted data points) will vary. Moreover, there are certain analyses (e.g. in **Figure 5**, previously **Figure 4**) that include only subjects with more than 2 good MRI sessions. By examining the degrees of freedom, the reader can see which measures have missing data points. In our previous submission, we were careful to note whenever we included additional covariates, e.g., subject motion or baseline reading skill. We have further revised the text in hopes of making it clear whenever subjects are excluded (i.e., for having usable data for only a single time point). Critically, we replicate all of the main effects in the paper using the subset of subjects with all available tests, and we replicate our intervention driven effects using an even more stringent motion cutoff threshold (0.3 vs. 0.7). Moreover, we have adopted modeling approaches (e.g., linear mixed effects models, rather than repeated measures ANOVA) that elegantly handle missing data.

-For Fig4 why are only 20 subjects shown?

-Why are the degrees of freedom so high for the behavioral measures (77 from 24 subjects pre and post)? Also what specific test is this tscore referring to? Since it is a linear mixed model I would expect an Ftest, unless you are testing a specific contrast of model coefficients.

We initially framed this section of the Results around the pre-post contrast; we have revised and reorganized the results and now report the omnibus F statistic associated with the model using intervention time (in hours). We agree that this streamlined presentation of the behavioral results is clearer.

And why are the DoF different across tests (77 for WJ basic reading and TOWRE, but 76 for WJ reading fluency, 63 for WJ calculation, and 68 for WJ math fact fluency)? If subjects were excluded for motion during MRI, please justify excluding them from behavioral measures as well.

As noted above, this reflects the different number of available data points for each of these tests. By examining the degrees of freedom, the reader can see which measures have missing data points. Critically, the effects reported in the manuscript do not depend on using the full data set vs. the subset of subjects with all available tests.

-The DoF reported for the findings of Fig1 are (3,76) which implies 80 observations across the 4 sessions. You claim that of the 93 intervention data points from the intervention group, and you excluded 9, leaving 84 observations. Please explain, considering that this works out for the reported control data (52 sessions excluding 3 = 49, and reported DoF is (3,45) = 49 observations).

We have 93 intervention data points, of which 13 are excluded from the diffusion data set (e.g., due to subject motion). We have updated the Methods section to reflect this.

-Please explain the DoF for each test, accounting for all covariates, number of subjects and data points included.

We hope that it is clear throughout the text whenever a covariate is included in a given model. We note that all subject and data exclusion criteria are all implemented in our analysis code, and do not reflect arbitrary decisions. Once the manuscript and code are published, a reader will be able to see how the full dataset gets transformed into each statistic and figure.

-Why only 16 filled dots in the large scatter plots of Fig2? FigS1 makes it clear that there should be 22 subjects with data from session 1. Why are the p values in Fig2 consistent with Pearson correlations from 20 subjects?

There are 20 subjects with usable reading and diffusion MRI data from Session 1. **Figure 1** (previously **Figure 2A**) should be consistent with this.

What data were used to for the PCA for “reading skill”? Which groups were included and what time points? It should be performed on the full dataset before intervention, and the transform then applied to each subsequent session so that it is not biased towards variability in each session for which the resulting PCA scores will be used. What are the resulting PCA coefficients

so the reader can know which measure contribute most? Indeed it is a little strange to produce a generalized score based on the hyperplane that contains the highest variance across your subjects, but since it correlates so well with the composite scores it seems legitimate. Please also include all analyses that use “reading skill” but instead using each of the WJ and TOWRE metrics instead. The authors state that they perform this analysis to avoid multiple comparison issues, but since it is only 4 metrics, it truly does not impact a bonferroni correction by very much at all.

—The supplementary results included for reviewer 2, issue 3 is not a satisfactory response. There are many more analyses in the text that use “reading skill”. Please perform all analyses with each reading index.

Using PCA or factor analysis to create a composite measure is standard practice both in reading research, and more broadly in the behavioral sciences. The benefit of creating a composite measure, rather than analyzing each individual sub-test, is that the composite is much more reliable (higher SNR) than any of the individual measures. It also cuts down on multiple comparisons and the desire to interpret small differences between different behavioral tests. Since the composite is a linear combination of the individual variables, it is a parsimonious way to summarize the major effects in the data. In response to Reviewer #2, we have included a series of additional tables and supplementary figures showing the results across different indices of reading. Reviewer #2 considered our response satisfactory, and we agree that adding these additional details added more information that will be of interest to reading researchers.

I strongly agree with previous reviewer 1, issue 4. An enormous interest in this manuscript hinges on individual subject results. Can you predict effectiveness of intervention from session 1 data? Do changes in tract metrics predict changes in reading ability for an individual subject? You have 21 subjects with 4 time-points, and so this is unquestionably feasible. I do not accept that this should be “salami-chopped” into a separate publication and is certainly more appropriate for this manuscript than the analyses in Fig5. Even a leave-one-subject-out cross validation scheme for predicting individual subject results should be included if at all possible, and would immensely strengthen this paper.

There is no doubt that identifying a neural biomarker for predicting children’s response to intervention is an extremely high-impact endeavor, both scientifically, and in terms of relevance to society. However, we are uncomfortable with the suggestion to disregard best principles of reproducible science, and report the results of an underpowered analysis. The dataset was not designed for this purpose and, based on a power analysis, we do not have sufficient power to reliably detect this effect. If we assume a moderate correlation of 0.5 between pre-intervention diffusion measures and subsequent growth, than we have a 65% chance of detecting a significant effect at $p < 0.05$. Given the low power of this analysis, any effect we detect should be considered exploratory, and would support the worthiness of pursuing a larger study targeting this question, but would not be appropriate for publication. We are okay reporting some exploratory analyses but, given the impact of proposing a biomarker for intervention efficacy, we are not comfortable reporting the results of this analysis.

Despite our reluctance to include the biomarker analysis in the paper we have performed this analysis for the reviewer. There are some effects. The figure below shows a correlation between baseline diffusivity and intervention-driven change in reading scores. The rendering of the pathways is colored based on the correlation at each point. There are some points on the tract where the correlation reaches statistical significance, but the correlation is also highly variable along the tract profile. Our reason for not performing this analysis in the paper is that we would like to adhere to guidelines for reproducible research laid out in recent articles such as Munafo and colleagues (A manifesto for reproducible science, *Nat Hum Beh*) and Button and colleagues (Power failure: why small sample size undermines the reliability of neuroscience, *Nat Rev Neurosci*). This is not an attempt to “salami chop” this into a subsequent paper: once the data is made public any other researcher can explore and publish biomarkers from this dataset as they deem appropriate. And since the response letter is published alongside the manuscript, readers will be able to appreciate that this sort of prediction is feasible.

The authors state in Methods that mean, radial, and axial diffusivity (MD, RD, AD) and fractional anisotropy (FA) were mapped onto each tract to create a “Tract Profile”. Why are these metrics not reported when a) they must have been computed in order to produce FA and MD, and b) they have been previously reported to relate to reading ability (e.g. Yeatman et al., 2011) and dyslexia (e.g. Vandermosten et al., 2012)?

There are many ways to summarize the shape of the diffusion tensor. We chose to analyze and report FA and MD because they are mathematically orthogonal metrics describing the tensor, and they are commonly reported, permitting us to contextualize our pre-intervention results in a larger body of literature. But we could also recompute each result using radial diffusivity, axial diffusivity, planarity, linearity, mode, or other shape statistics. For completeness, we include an analysis of intervention driven change in both axial and radial diffusivity in **Supplementary Figure 6**. Because these analyses do not provide substantial novel information, we prefer to

focus on MD and FA as the manuscript already has 11 Figures and 11 Tables between the main text and supplement.

Minor corrections:

FigS1a should say RMS displacement < (not >) 0.7mm

FigS2 - *between (sp.)

FigS4 - *reading (sp.)

Thank you for the corrections. We have made these adjustments.

Reviewers' comments:

Reviewer #1 (Remarks to the Author):

This manuscript explores the change in white matter as children gain reading intervention. It certainly adds to our understanding of stable and dynamic changes in white matter as an individual learns to read. The manuscript has been substantially revised as a result of reviewer feedback. Each analysis is now better motivated and the figures are well-explained. The change in subject number for the different analyses is also addressed as a result of feedback from reviewers 2 and 4. The online release of the data is potentially helpful as well. I have a few small questions that I hope the authors can answer:

For the typical readers, is there a session-to-session reliability measure of white matter measures for all 18 tracts, given that the changes across sessions are happening globally? Is there a similar Fig. 3 but for typical readers? The se seems quite large for the typical readers, and it would be interesting to see the session-to-session differences for that group.

Fig.3: It seems that reading intervention does help 'normalize' reading skill somewhat. If the changes in white matter do not explain this gain in reading skill, what neural measure does? This is a critical point that needs to be addressed.

Reviewer #2 (Remarks to the Author):

The authors have been very responsive and the changes to the manuscript offer significant improvement and clarity. There are just a few issues that remain.

1. It seems the authors may have misunderstood the first point in the previous review. The suggestion was to describe in the results section the intervention effects first (behavioral then neural), followed by individual differences. In that way all the group intervention effects are together, and all the individual differences findings are together.

2. The authors now include an ANOVA on the behavioral data, as requested. However, it's perplexing why this was limited to the reading-matched control group (n=9) instead of the whole group, especially since the imaging results are all based on the whole control group. The reading-matched group could be included in the supplemental materials or in addition to the whole group, but the main text should include the description of the ANOVA for the whole control group (n = 19). Only in this way will the description of the behavioral results mirror the imaging results.

3. Figure 3, the authors could add a dot (mean) and lines (SE) to more clearly illustrate that the gray circle/ellipse of typical readers is illustrating this information.

4. I don't think the authors fully responded to the first part of Point 6. Could they provide some description in the discussion section about why the brain-behavior correlations may be changing with experience?

We thank both reviewers for their positive feedback on the revised manuscript. Both have provided a few additional suggestions this round, which we have addressed in our revision. Our point-by-point response follows. Additionally, we have changed the manuscript title to conform to Nature Communications formatting guidelines.

Reviewer #1 (Remarks to the Author):

This manuscript explores the change in white matter as children gain reading intervention. It certainly adds to our understanding of stable and dynamic changes in white matter as an individual learns to read. The manuscript has been substantially revised as a result of reviewer feedback. Each analysis is now better motivated and the figures are well-explained. The change in subject number for the different analyses is also addressed as a result of feedback from reviewers 2 and 4. The online release of the data is potentially helpful as well.

I have a few small questions that I hope the authors can answer:

For the typical readers, is there a session-to-session reliability measure of white matter measures for all 18 tracts, given that the changes across sessions are happening globally?

We have quantified the reliability for each tract in the control group by calculating Pearson's r across sessions. In control subjects, the median reliability across tracts was $r = 0.73$ for mean diffusivity and $r = 0.76$ for fractional anisotropy. Reliability estimates for all tracts and both parameters are now provided in **Supplementary Table 11**.

Is there a similar Fig. 3 but for typical readers? The se seems quite large for the typical readers, and it would be interesting to see the session-to-session differences for that group.

Figure 3 shows the Session 1 data for typical readers. The SE is relatively large because our control group includes subjects with a range of initial reading levels (e.g., scores on the WJ Basic Reading Skills composite at Session 1 range from 87 to 121).

We now include a new supplementary figure (**Figure S7**) that reproduces the intervention data alongside the same mean and SE plots for the full sample of control subjects, so that the reader can more easily assess session-to-session stability in the control group. Because many of the control subjects completed fewer than 4 sessions, plots of the group means at each timepoint understate the stability of individual subjects. This difference in sample size is accounted for in our statistical analysis. Nonetheless, this visualization makes clear that the control group does not show the systematic shifts in either behavior or white matter properties that are seen over the intervention period. This is in line with the results of our statistical analyses.

Figure S7. Reading scores and white matter tissue properties do not change systematically in the control group. Intervention data are plotted alongside the session-to-session means and standard errors for the full sample of control subjects. Because many of the control subjects completed fewer than 4 sessions, the control samples at each time point do not overlap perfectly. We omit session 4 due to the small number of available subjects. Session 1-3 data are shown in dark to lighter gray. Color-coding for the intervention subjects is the same as Figure 3, with brighter colors corresponding to later time points. Intervention subjects show large changes in behavior and white matter tissue properties between sessions 1, 2 and 3, but control subjects do not. Group by session interactions are reported in the main text of the Results.

Fig.3: It seems that reading intervention does help ‘normalize’ reading skill somewhat. If the changes in white matter do not explain this gain in reading skill, what neural measure does? This is a critical point that needs to be addressed.

Within the intervention group, changes in the white matter account for behavioral gains, as reported in the section of the Results titled: “White matter properties of the AF and ILF change in concert and track individual learning”. However, these changes do not ‘normalize’ white matter differences, in the sense that post-intervention subjects do not look more like their typically reading peers. This result is in line with the idea that growth in reading skill may be accomplished via compensatory mechanisms that differ from those supporting the acquisition of skilled reading in typical children. This interpretation is consistent with theories that have been developed in the functional magnetic resonance imaging (fMRI) literature on reading interventions.

We now note in the Results and Discussion that an outstanding question in the intervention literature is the extent to which behavioral remediation involves normalization of neural deficits, versus the development of different compensatory mechanisms. In the extensive fMRI literature, there are some findings that show normalization of brain function, and others that suggest compensatory mechanisms in people with dyslexia. Our study is among the first to examine this topic from the perspective of intervention induced change in the white matter. While normalization and compensation are not mutually exclusive, our data is more consistent with a compensatory mechanism.

We have updated the following paragraph in the Results to introduce and cite the literature on compensatory mechanisms:

One possible interpretation of group differences in MD and FA between good and poor readers is that these differences reflect abnormal tissue properties in poor readers. In that case, one might predict that remediation of reading difficulties would involve a ‘normalization’ of deficits in white matter structure. Alternatively, plasticity in the white matter might reflect a compensatory mechanism that supports the learning process. In that case, case white matter tissue properties in the remediated readers would not necessarily look more similar to those in the typical reading control subjects.

And we now elaborate on this idea in the Discussion section:

Within the intervention group, changes in the white matter are linked to changes in behavior (**Figure 5**). However, these effects do not follow the trajectory predicted by a normalization account, in which remediation of reading difficulties could be expected to eliminate differences in the white matter between children with dyslexia and typical readers. A number of functional imaging studies have previously examined the extent to which behavioral remediation involves normalization of neural deficits, versus development of different or compensatory mechanisms⁶⁴. Several studies have reported normalization of responses after successful remediation⁷⁸⁻⁸³, but many studies also highlight neural changes in regions that are not considered fundamental components of the circuit in skilled readers. These learning-induced changes have been interpreted as reflecting compensatory responses^{80,81,83,84}. Our results are compatible with the idea that remediation may be accomplished through compensatory mechanisms that differ from those supporting the acquisition of skilled reading in typical children.

Reviewer #2 (Remarks to the Author):

The authors have been very responsive and the changes to the manuscript offer significant improvement and clarity.

There are just a few issues that remain.

1. It seems the authors may have misunderstood the first point in the previous review. The suggestion was to describe in the results section the intervention effects first (behavioral then neural), followed by individual differences. In that way all the group intervention effects are together, and all the individual differences findings are together.

We currently start by presenting a replication of previously reported correlations between white matter and reading skills, prior to the intervention. We then discuss intervention effects, followed by an examination individual differences over the course of intervention. We feel that this organization is important to set the stage for the rest of the findings and we prefer not to move this figure back to a later place in the manuscript. However, if the reviewer feels strongly that this replication is out of place and detracts from the flow of the paper, we can move these results to a supplementary figure.

2. The authors now include an ANOVA on the behavioral data, as requested. However, it's perplexing why this was limited to the reading-matched control group (n=9) instead of the whole group, especially since the imaging results are all based on the whole control group. The reading-matched group could be included in the supplemental materials or in addition to the whole group, but the main text should include the description of the ANOVA for the whole control group (n = 19). Only in this way will the description of the behavioral results mirror the imaging results.

We have reorganized the relevant section so that the behavioral results for the full control group are presented first, mirroring the imaging results. We now report results for the reading matched control group in **Supplementary Table 2**.

3. Figure 3, the authors could add a dot (mean) and lines (SE) to more clearly illustrate that the gray circle/ellipse of typical readers is illustrating this information.

We have revised **Figure 3** as the reviewer suggests.

4. I don't think the authors fully responded to the first part of Point 6. Could they provide some description in the discussion section about why the brain-behavior correlations may be changing with experience?

To preserve the brain-behavior correlations present at session 1, the neural and behavioral changes would have to occur at the same rate, not only within individuals, but also *across* individuals. In addition, the intervention would have to 'normalize' the white matter: Individuals with lower reading scores would have to become more similar to individuals with higher reading scores, both in terms of brain and behavior. As we note above, and in the manuscript, we observe individual differences in the timing of intervention effects, and we also note that learning alters the white matter, but does not *normalize* it. Therefore, we suggest that it may be misleading to interpret brain-behavior correlations at a single time-point during learning.

We have added the following text to the Discussion to clarify these points and relate our interpretation to the broader literature on dyslexia:

Given the magnitude of short-term, learning-induced changes in the white matter, previously reported group differences may be driven in part by environmental differences between groups, since systematic difference in the environment (e.g., differences in the quality or intensity of recent educational experiences for dyslexic versus control subjects) could be expected to exert a large influence on diffusion

measurements, and to potentially counteract or change pre-existing anatomy-behavior relationships. This offers an explanation for why some studies find a positive correlation between FA and reading skills^{14,19}, while other studies find a negative correlation between FA and reading skills^{8,85} within the exact same tracts. In a single group of subjects, we find that measured anatomy-behavior correlations change over the course of learning. Since learning effects do not ‘normalize’ the white matter, differences in the timing of learning effects among subjects naturally lead to changes in the correlations computed at any given time point. An important implication of the present work is that anatomy-behavior correlations taken at a single time-point should be interpreted cautiously, so long as the relevant behaviors and anatomical properties are subject to experience-dependent change.

REVIEWERS' COMMENTS:

Reviewer #1 (Remarks to the Author):

The authors have addressed my concerns and I have no further suggestions.

Reviewer #2 (Remarks to the Author):

The authors have addressed all of the remaining concerns in this revised version of the manuscript.